# A genome-wide atlas of antibiotic susceptibility targets and pathways to tolerance

Dmitry Leshchiner[1,5], Federico Rosconi [1,5], Bharathi Sundaresh [1,5], Emily Rudmann[1,5], Luisa Maria Nieto Ramirez [1,5], Andrew T. Nishimoto [2], Stephen J. Wood [1], Bimal Jana[1], Noemí Buján [1], Kaicheng Li[3], Jianmin Gao [3], Matthew Frank[2], Stephanie M. Reeve [4], Richard E. Lee [4], Charles O. Rock [2], Jason W. Rosch [2] & Tim van Opijnen [1✉]

Detailed knowledge on how bacteria evade antibiotics and eventually develop resistance could open avenues for novel therapeutics and diagnostics. It is thereby key to develop a comprehensive genome-wide understanding of how bacteria process antibiotic stress, and how modulation of the involved processes affects their ability to overcome said stress. Here we undertake a comprehensive genetic analysis of how the human pathogen *Streptococcus pneumoniae* responds to 20 antibiotics. We build a genome-wide atlas of drug susceptibility determinants and generated a genetic interaction network that connects cellular processes and genes of unknown function, which we show can be used as therapeutic targets. Pathway analysis reveals a genome-wide atlas of cellular processes that can make a bacterium less susceptible, and often tolerant, in an antibiotic specific manner. Importantly, modulation of these processes confers fitness benefits during active infections under antibiotic selection. Moreover, screening of sequenced clinical isolates demonstrates that mutations in genes that decrease antibiotic sensitivity and increase tolerance readily evolve and are frequently associated with resistant strains, indicating such mutations could be harbingers for the emergence of antibiotic resistance.

[1] Biology Department, Boston College, Chestnut Hill, MA 02467, USA. [2] Department of Infectious Diseases, St. Jude Children's Research Hospital, Memphis, TN 38105, USA. [3] Chemistry Department, Boston College, Chestnut Hill, MA 02467, USA. [4] Department of Chemical Biology and Therapeutics, St. Jude Children's Research Hospital, Memphis, TN 38105, USA. [5]These authors contributed equally: Dmitry Leshchiner, Federico Rosconi, Bharathi Sundaresh, Emily Rudmann, Luisa Maria Nieto Ramirez. ✉email: vanopijn@bc.edu

The emergence of antibiotic resistance in bacterial pathogens is a continuously developing complex problem that is only solvable if besides new drugs we also learn to understand the exact (genetic) processes that enable resistance. For instance, new antibiotics and treatment strategies are key to retain the ability to treat resistant infections. However, a comprehensive understanding of how and under which conditions resistance emerges, which genes and pathways contribute to drug sensitivity, and how resistance may be prevented or even taken advantage of, are equally important, as it could make treatments more focused and possibly less dependent on new drugs. For many antibiotics, we know which genomic changes can cause resistance. However, it is often not clear how we get there with respect to which evolutionary paths are taken and whether for instance tolerance or lowered drug sensitivity precedes resistance. Interestingly, clinical strains isolated during antibiotic treatment failure may lack known resistance markers and instead contain multiple changes that may have no clear or known role in resistance[1–5]. However, whether these changes play a role or not is often unclear because the distribution of changes that can affect a bacterium's drug sensitivity are largely unknown[1–7]. Therefore, understanding which genes, pathways and processes can contribute to altered drug susceptibility, could help identify genomic changes that not only sensitize bacteria to certain drugs, but desensitize them and may thereby act as precursors for antibiotic escape and/or resistance development.

Resistance emerges primarily through drug target mutations blocking antibiotic lethal action, upregulation of efflux pumps, and the acquisition of drug inactivating enzymes[7–13]. Importantly, an antibiotic's effects go far beyond the interaction with its direct target. We, and others, have shown that when a bacterium is challenged by an antibiotic, the imposing stress can expand throughout the bacterium and affect and demand the involvement of many different processes[6,14–17]. For instance, while fluoroquinolones like ciprofloxacin inhibit DNA replication by targeting gyrase and/or topoisomerase, this also triggers double-stranded breaks requiring the involvement of DNA repair mechanisms, which in turn requires nucleotide and energy metabolism. Antibiotics can thereby trigger a stress cascade, that with mounting stress increasingly reverberates through the organismal network, until the accumulating stress passes a threshold at which point the organism succumbs to the pressure[15,17]. This explains why mutations in genes or pathways involved in dealing with the downstream (indirect) effects of antibiotic exposure can often make a bacterium more sensitive to a specific antibiotic. Indeed, we have shown for *Streptococcus pneumoniae* and *Acinetobacter baumannii* that, for instance, targeting DNA repair makes bacteria more susceptible to DNA synthesis inhibitors (DSIs)[6,16,18], or targeting the Rod-system and/or Divisome makes *A. baumannii* more sensitive to cell-wall synthesis inhibitors (CWSIs)[6]. This means that downstream genes, pathways and processes can be used as new targets or drug potentiators, either by themselves or in combination with others[6,14]. Moreover, in most bacteria, as in any other organism, the majority of genes are of unknown function, it is unclear what role they play in a specific process and/or pathway, or how they are connected within the organismal genomic network. Thus, besides solving gene function, mapping-out which genes, pathways and processes are involved in dealing with and overcoming antibiotic stress, and how they interact with each other, can provide key insights into uncovering new drug targets, or for instance rational combination strategies[6].

While identifying off-target genes and pathways that increase drug sensitivity may thus be useful, it is possible that changes in associated processes could, in contrast, just as well reduce the experienced antibiotic stress. Such changes would thereby decrease antibiotic sensitivity and could possibly function as precursors to the emergence of resistance. A possible example of this is tolerance and/or persistence, where a small proportion of cells in a population can be induced by external conditions including nutrient starvation[19], cell density[20], antibiotic stress[21] and stress from the immune system[22] into a cell state that enables them to tolerate high (transient) concentrations of antibiotics. Cell states associated with tolerance include cell dormancy, slow growth, transient expression of efflux pumps, and induction of stress response pathways[23–26]. However, the mechanistic underpinnings of tolerance and decreased antibiotic sensitivity remain largely undefined and possibly differ between bacterial species and vary among antibiotics[27]. Moreover, specific mutations can (dramatically) increase the fraction of the surviving population[28–30], indicating these tolerant phenotypes have a genetic basis. Lastly, since clinical isolates often carry mutations located outside well-characterized drug targets[1–5,31,32], they could thus be composed of variants with different antibiotic sensitivities. Consequently, such variants with decreased antibiotic sensitivity could enable antibiotic escape, and/or enable multi-step high-level resistance mutations to evolve as they are given an extended opportunity to emerge[25,33–36]. Variants with decreased antibiotic sensitivity may thereby play an important role in antibiotic treatment failure[5,37,38]. However, the breadth of possible genetic alterations that can enable (increased) tolerance and/or decrease antibiotic sensitivity are largely unknown, making it unclear how often and probable it is that such variants arise.

In this study, we use Tn-Seq in *S. pneumoniae* exposed to 20 antibiotics, 17 additional environments, and two in vivo infection conditions, to generate a genome-wide atlas of drug susceptibility determinants and build a genome-wide interaction network that connects cellular processes and genes of unknown function. We explore several interactions as new leads for gene function, while we show that specific interactions can be used to guide the identification of targets for new antimicrobial strategies. We highlight one such novel target in the membrane, by successfully developing a combinatorial antibiotic-antibody strategy that significantly reduces the bacterial load during an acute mouse lung infection. Furthermore, detailed mapping of antibiotic sensitivity data to pathways and genes with known function suggests a multitude of genome-wide genomic changes exist that can make the bacterium less susceptible and often tolerant to specific antibiotics. We untangle some of the underlying genetic mechanisms and show that decreased susceptibility and tolerance can come from a variety of changes including those in (nucleotide) metabolism, (p)ppGpp and ATP synthesis, transcription, and translation, as well as different types of transport. By further combining in vivo-infection- with antibiotic-Tn-Seq data we predict and experimentally validate that many disruptions may retain their decreased antibiotic sensitivity phenotype in vivo, and thereby outcompete the wild type in the presence of antibiotics. Moreover, by screening hundreds of clinical isolates we show that changes in genes that can decrease antibiotic sensitivity readily evolve in human patients and are often associated with antibiotic resistance. Consequently, these data highlight the wide array of possibilities that can lead to lowered antibiotic sensitivity and/or tolerance and underscore the importance of understanding the genetics of variants with altered drug susceptibility.

## Results

**A genome-wide view of antibiotic sensitivity.** To obtain a genome-wide view of the genetic determinants that can modulate antibiotic stress in *S. pneumoniae*, Tn-Seq was employed in the presence of 20 antibiotics (ABXs), representing 9 different ABX groups and four classes including cell wall synthesis inhibitors

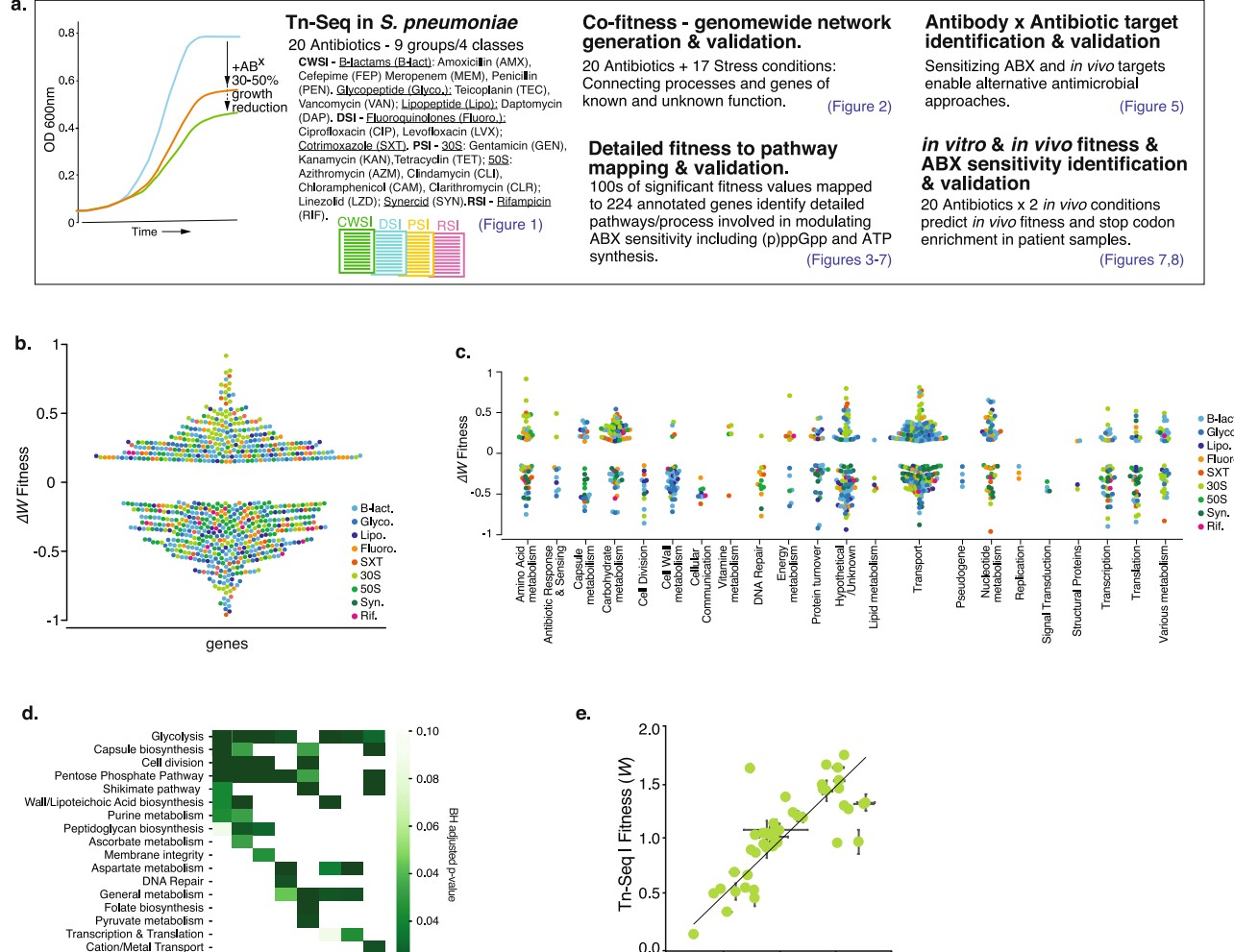

**Fig. 1 A genome-wide atlas of negative and positive fitness effects, highlights a multitude of processes that can modulate antibiotic susceptibility.** **a** Project setup and overview. Tn-Seq is performed with *S. pneumoniae* TIGR4, which is exposed to 20 antibiotics at a concentration that reduces growth by 30–50%. Genome-wide fitness is determined for each condition, suggesting a multitude of options exists to increase as well as decrease antibiotic sensitivity. A co-fitness network is constructed by adding Tn-Seq data from 17 additional conditions, which through a SAFE analysis highlights functional clusters, and connects known and unknown processes. The genome-wide atlas and network are used to develop an antibiotic-antibody combination strategy, and to map out the wide-ranging options that can lead to decreased antibiotic sensitivity in vitro and in vivo and that are associated with a higher rate of stop codons in clinical samples. **b** There are a large number of genetic options that can modulate antibiotic sensitivity; with significant increased ($\Delta W < -0.15$) and decreased sensitivity ($\Delta W > 0.15$) split over all antibiotics almost equally likely. **c** Additionally, increased and decreased antibiotic sensitivity are distributed across a wide variety of functional categories. **d** Enrichment analysis shows that some pathways/processes such as glycolysis are relatively often involved in modulating responses to antibiotics, while other processes are more specific. **e** Validated growth experiments ($n \geq 3$ independent experiments) performed throughout the project highlight the Tn-Seq data is of high quality. ±SEM are shown for each data point. Source data are provided as a Source Data file.

(CWSIs), DNA synthesis inhibitors (DSIs), 30 S and 50 S protein synthesis inhibitors (PSIs) and an RNA synthesis inhibitor (RSI) (Fig. 1a). Six independent transposon libraries were generated and grown for approximately 8 generations in the absence and presence of an antibiotic at a concentration that reduces growth by approximately 30–50% (Supplementary Data 1). Tn-mutant frequencies are determined through Illumina sequencing from the beginning and end of the experiment with high reproducibility between libraries ($R^2 = 0.70$–$0.90$; Supplementary Fig. 1) which is consistent with previous Tn-Seq experiments[6,15,16,18,39–42]. Combined with the population expansion during the experiment each mutant's fitness ($W_{MT}$) is calculated to represent their environment-specific relative growth rate, which means that a mutant with for instance a fitness of 0.5 ($W_{MT} = 0.5$) grows twice

as slow as the wild type (WT)[6,18,39,43,44]. Each gene's antibiotic-specific fitness is statistically compared to baseline fitness without ABXs, and is represented as $\Delta W$ ($W_{ABX} - W_{noABX}$) and categorized as: (1) Neutral, $\Delta W = 0$, a mutant's relative growth is similar in the absence and presence of an ABX; (2) Negative, $\Delta W < 0$, a mutant's fitness is significantly lower and thus grows relatively slower in the presence of an ABX; (3) Positive, $\Delta W > 0$, a mutant's fitness is significantly higher and thus grows relatively faster in the presence of an ABX. All antibiotics trigger both positive and negative fitness effects (Fig. 1b, Supplementary Data 2), which are distributed across 22 different gene categories (Fig. 1c). Importantly, enrichment analysis shows there are multiple expected patterns, for instance genes involved in DNA repair are enriched in the presence of fluoroquinolones; cell-wall, peptidoglycan, and

cell-division genes are enriched in ß-lactams and glycopeptides; membrane integrity genes in lipopeptides; and transcription and translation in PSIs (Fig. 1d). Additionally, throughout the manuscript, we validate a total of 49 predicted genotype × phenotype interactions, which indicates the Tn-Seq data is of high quality and in line with previously shown accuracy[6,15,16,18,39–42] (Fig. 1e, Supplementary Data 8).

**Co-fitness interaction networks identify known and unknown genetic relationships.** Screens such as Tn-Seq are geared toward highlighting the genetic regions and/or genes that are important under a specific screening condition. With increasing conditions, genes acquire profiles that reflect their involvement/importance in those conditions, where genes with similar profiles indicate having similar and/or shared tasks. Such profiles can thereby help fill gaps in pathways, and/or identify genes and gene clusters with similar roles. By building a correlation matrix based on each gene's ABX fitness-profile patterns emerge along a similarity range; from genes with highly similar to contrasting profiles. Moreover, to increase statistical power (i.e., more conditions increases the ability to identify more and stronger associations) the ABX dataset was supplemented with previously collected Tn-Seq data from 17 additional non-antibiotic conditions[18] (Supplementary Data 3). This results in a $1519 \times 1519$ gene matrix where positive correlations between genes come from shared phenotypes (i.e., similar profiles), while negative correlations come from opposing phenotypic responses under the same condition (i.e., contrasting profiles; Supplementary Data 4). By repeatedly hiding random parts of the data the stability and strength of each correlation is calculated and represented in a stability score (Supplementary Data 5). The correlation matrix and stability score are turned into a network, where each node is a gene, and each edge is a correlation coefficient above a threshold (>0.75), which combined with the stability score indicates the strength of the relationship between two genes. (Fig. 2a; Supplementary Data 6). Spatial Analysis of Functional Enrichment (SAFE)[45,46] is used to define local neighborhoods within the network, i.e., areas enriched for a specific attribute (e.g., a pathway or functional category), which identifies multiple clusters that represent specific pathways and processes including purine metabolism, cell-wall metabolism, cell division and DNA repair (Fig. 2b; Supplementary Data 7). Moreover, the network contains gene clusters of high connectivity identifying highly related genes including those within the same operon such as the *ami*-operon, an oligopeptide transporter, the *dlt*-operon which decorates wall and lipoteichoic acids with d-alanine, and the *pst*-operon a phosphate transporter (Fig. 2c, I–III). Besides identifying known relationships, the network also uncovers interaction clusters between genes with known and unknown interactions and function. Several such clusters are highlighted in Fig. 2c (IV–VIII), including genes involved in purine metabolism (further explored below), threonine metabolism, and in secretion of serine-rich repeat proteins (SRRPs), which are important for biofilm formation and virulence[47]. Importantly, the identification of biologically relevant relationships among (clusters of) genes indicates the data is rich in known and new information.

**Detailed pathway mapping identifies processes that simultaneously increase and decrease drug susceptibility in an antibiotic-specific manner.** Two hundred and twenty-four genes with a known annotation are present in the data that have at least one significant phenotype in response to an antibiotic, which can be split over 21 functional groups according to a pathway or process they belong to (Fig. 3a). Each group is characterized by having multiple instances of decreased fitness, indicating genes

that upon disruption increase sensitivity to one or more antibiotics (negative phenotype). Additionally, each group, except for cell division, also has multiple instances that increase fitness, which is suggestive of genes that upon disruption decrease antibiotic sensitivity (Fig. 3a; positive phenotype). Moreover, each antibiotic group triggers both negative and positive effects (Fig. 3b). Where possible, the 21 functional groups are organized according to a pathway or process they belong to and each gene is combined with its antibiotic susceptibility profile. This results in an antibiotic susceptibility atlas, which shows on a fine-grained scale, how inhibiting a pathway or process can seemingly simultaneously lead to increased and decreased drug susceptibility in an antibiotic-specific manner (Fig. 3c and Supplementary Figs. 2 and 3). For instance, in the glycolysis group, knocking out any of the three genes involved in forming the phosphotransferase (PTS)-system (SP_0282-SP_0284) that imports glucose to generate glucose-6-phosphate (G-6P), has a negative effect on fitness in the presence of 30S and 50S PSIs as well as Synercid (a synergistic combination of two PSIs), while it increases fitness in the presence of all CWSIs (ß-lactams, glycopeptides, and daptomycin) and fluoroquinolones. Also, knocking out SP_0668 (*gki*, glucokinase), an enzyme that converts α-D-Glucose into G-6P, has a positive effect on fitness in all CWSIs and a negative effect in 30S PSIs. In contrast, inhibiting SP_1498 (*pgm*, phosphoglucomutase), the major interconversion enzyme of G-6P and G-1P, has a negative effect on fitness with all antibiotics (Fig. 3c). Additional detailed examples are highlighted in Fig. 3c, for instance for pyruvate metabolism, where inhibiting lactate, or acetaldehyde production increases sensitivity to ß-lactams and glycopeptides and decreases sensitivity to 30S PSIs, inhibiting formate production decreases sensitivity to co-trimoxazole and 30S PSIs, and inhibiting acetyl-phosphate production decreases sensitivity to ß-lactams, glycopeptides, and co-trimoxazole. Within aspartate metabolism a range of changes can be triggered from increased sensitivity to ß-lactams, and glycopeptides, to decreased sensitivity to most other antibiotics. Moreover, the four genes involved in the production of threonine from L-aspartate trigger decreased sensitivity to fluoroquinolones and 30S and 50S PSIs. In the shikimate pathway inhibiting the production of chorismate from phosphoenolpyruvate (PEP) and erythrose-5-phosphate leads to increased sensitivity to ß-lactams, co-trimoxazole, and Synercid. Cell division is the only process that upon interference, only generates increased sensitivity, specifically to CWSIs and co-trimoxazole. Interfering with peptidoglycan synthesis also mostly leads to increased sensitivity to CWSIs, as well as to 30 S PSIs, while changes to genes that are involved in anchoring proteins to the cell wall (SP_1218 [*srtA*], SP_1833) can decrease sensitivity to CWSIs. Lastly, interfering with protein turnover, for instance through the protease complex ClpCP (SP_2194, SP_0746) and the regulator CtsR (SP_2195), which are generally assumed to be fundamental for responding to stress[48,49], leads to decreased CWSI sensitivity and increased sensitivity to 30S and 50S PSIs (Fig. 3c and Supplementary Fig. 2). Moreover, FtsH (SP_0013), important for clean-up of misfolded proteins from the cell wall, increases sensitivity to 30S PSIs and Synercid, indicating how important protein turnover is especially for surviving 30S PSIs, which can trigger the production of faulty proteins. Most importantly, these data show that, as expected, hundreds of options exist where disruption of a pathway or process leads to increased sensitivity to specific antibiotics. Remarkably, there seem to be almost as many options that can lead to decreased antibiotic sensitivity.

***cozEb* encodes a cell division and peptidoglycan synthesis embedded membrane protein that can be critically targeted in vivo through an antibody-antibiotic strategy.** By identifying

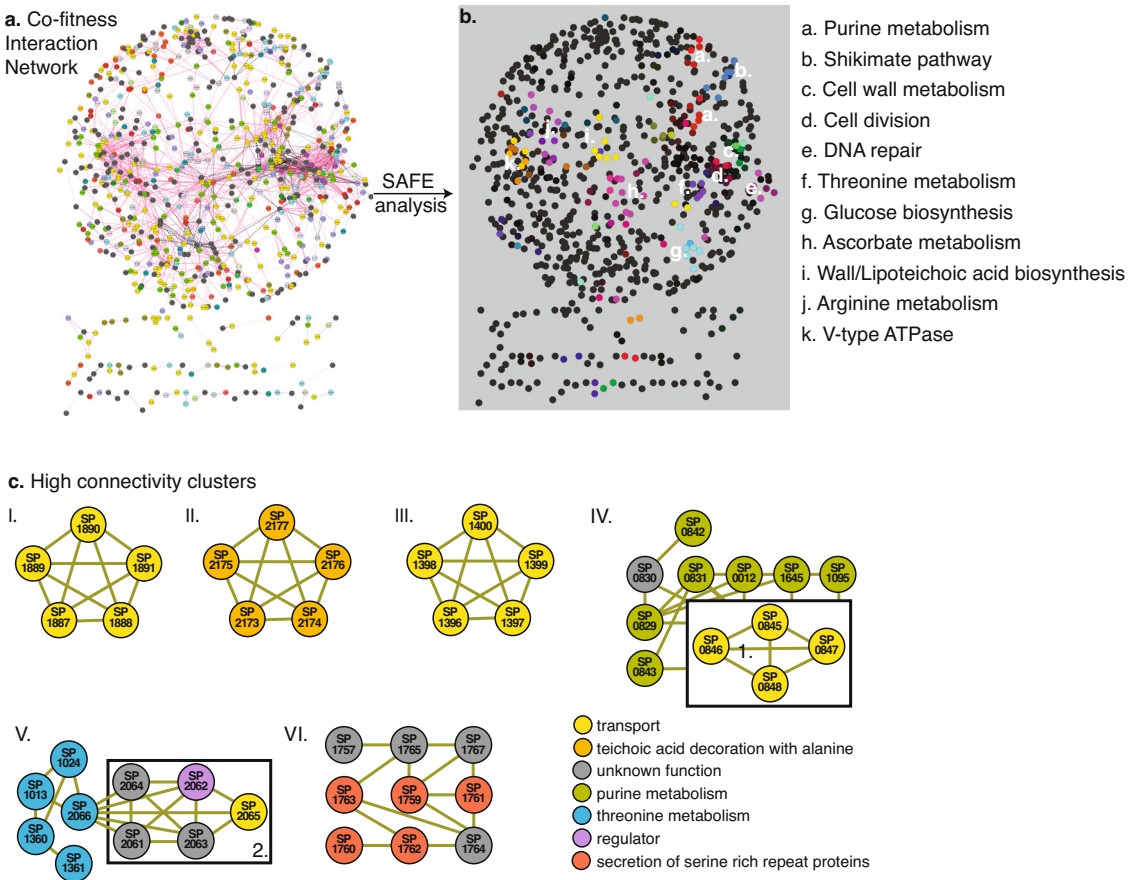

**Fig. 2 A co-fitness network identifies tight genetic clusters of known and unknown genes and processes. a** A 1519 × 1519 gene correlation matrix based on Tn-Seq data from 37 conditions generates a network with genes as nodes, and edges as interactions with a stability score and thresholded correlation >0.75. The network contains one large connected component and multiple smaller components placed underneath; **b** A SAFE analysis identifies at least 11 clusters within the network that represent specific pathways and processes; **c** The network contains highly connected clusters of smaller groups of genes for instance those within the same operon such as cluster: I. the *ami*-operon (unknown transport); II. the *dlt*-operon; and III. the *pst*-operon (phosphate transport). Several additional clusters are highlighted containing annotated and unannotated genes, connected through known and unknown interactions including cluster: IV. containing genes involved in purine metabolism and a putative deoxyribose transporter (boxed 1.); V. containing genes involved in threonine metabolism and several genes located as neighbors to SP_2066/*thrC* with unclear functions (boxed 2), including a regulator (SP_2062) and a transporter (SP_2065); VI. containing genes involved in secretion of serine-rich repeat proteins. Source data are provided as a Source Data file.

targets that (re)sensitize bacteria against existing antibiotics, genome-wide antibiotic susceptibility data have the potential to guide the development of new antimicrobial strategies. One such strategy could be a combined therapeutic antibody-antibiotic approach; the antibody would target a gene product that is important for sensitivity to one or more antibiotics and ideally the product would be easily accessible for the antibody at the bacterial cell surface. To find suitable candidate targets, Tn-Seq data were filtered for gene products that, based on a known function or localization prediction, are likely to be present in the cell wall or membrane, and that when disrupted, increase sensitivity to one or more antibiotics. Moreover, it would likely be ideal if the gene is also important for survival in vivo. A strong candidate is SP_1505, which in the interaction network is most tightly linked to cell wall metabolism and cell-division genes (Fig. 4a). After we previously hypothesized that it may play a role in cell wall integrity[14], it was recently named *cozEb*, with a likely role in organizing peptidoglycan synthesis during cell division[50], which fits its interaction profile (Fig. 4a). Importantly, the antibiotic Tn-Seq data suggest that disruption creates increased sensitivity to vancomycin and rifampicin, while the product is critical in the presence of daptomycin, which was confirmed through individual growth curves (Fig. 4b). The protein has eight predicted membrane-spanning domains

(Fig. 4c), and in vivo Tn-Seq predicts it is important for survival in both the nasopharynx and lungs (Fig. 4a, Supplementary Data 2). The gene was cloned into an expression plasmid generating an ~30kD product (Fig. 4c), which was used to raise rabbit anti-CozEb antibodies, which were confirmed to be specific for the *cozEb* gene product (Fig. 4c). Potential antibody in vitro activity was determined through a bacterial survival assay in the absence and presence of antibodies and either vancomycin or daptomycin. Incubating bacteria with antibodies or daptomycin has no significant effect on bacterial survival, while vancomycin alone at the concentration used slightly reduces the number of surviving bacteria. Moreover, combining the antibody with either vancomycin or daptomycin further reduces the number of surviving bacteria in vitro compared to any agent individually (Fig. 4d). To assess whether the antibody-antibiotic approach works in vivo, mice were intranasally challenged with a bacterial inoculum either containing WT or ΔcozEb. Two additional sets of mice were challenged with WT and 8 h post infection they were either treated with daptomycin and control IgG antibody or with daptomycin and CozEb-specific antibody. Mice were sacrificed 24 h post infection, and bacteria in the lungs were enumerated. As predicted by the in vivo Tn-Seq data the *cozEb* knockout has a significantly lower fitness in the lungs highlighted by an up to

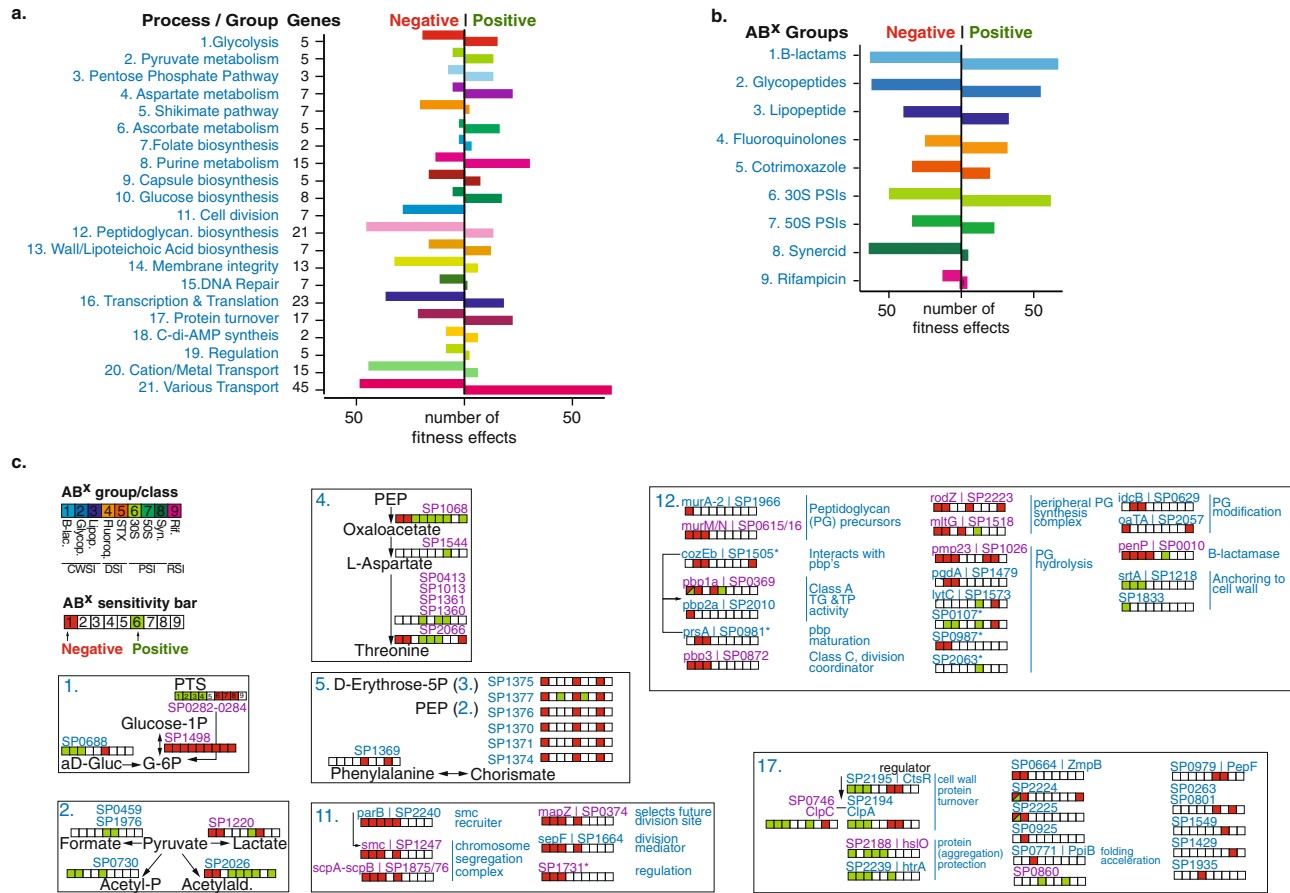

**Fig. 3 A multitude of options, pathways, and processes can simultaneously lead to increased and decreased antibiotic susceptibility. a** The number of phenotypes scored for each pathway/process. Genes with at least one significant phenotype are split over 21 groups according to a pathway or process they belong to, which highlights how modulation of most pathways can lead to increased (negative phenotype) and decreased (positive phenotype) antibiotic sensitivity. **b** The number of phenotypes scored for each antibiotic group. While sensitivity to each antibiotic (group) can be increased by knocking out genes in the genome (negative phenotype), sensitivity can be decreased (positive phenotype) almost as often for most ABXs, except for Synercid, and to a lesser extent rifampicin, where most effects are negative. **c** Detailed view of 7 out of 21 groups/processes highlighting how modulation of specific targets within each process leads to changes in antibiotic sensitivity. Each group is indicated with a number that is the same as in **a**. Where possible, genes are ordered according to their place in a process/pathway, and gene numbers (SP_) are combined with gene names and annotation. Each indicated gene is combined with an 'antibiotic sensitivity bar' indicating whether disruption leads to increased (red/negative fitness) or decreased (green/positive fitness) sensitivity to a specific or group of antibiotics. When phenotypic responses are the same, multiple genes are indicated with a single bar (e.g. SP0282/SP0283/SP0284 in glycolysis, or SP0413/SP1013/SP1361/SP1360 in Aspartate metabolism). Gene numbers in blue have no effect on growth in the absence of antibiotics when knocked out, while gene numbers in purple have a significant growth defect in the absence of ABXs (see for detailed fitness in the absence and presence of antibiotics Supplementary Data 2). Essential genes are not indicated and genes with an asterisk have a partial or tentative annotation that has not been resolved. All 21 groups are listed in Supplementary Figs 2 and 3. Source data are provided as a Source Data file.

2.5-log lower bacterial load compared to WT. Importantly, while the WT survives equally well in the presence of the low daptomycin concentration and the control IgG antibody, in the presence of daptomycin and the CozEb-targeting antibody, its survival in the lungs is significantly reduced and resembles that of the *cozEb* knockout (Fig. 4e). This shows that by combining antibiotic and in vivo Tn-Seq with gene annotation information, a gene product can be selected that is central and critical to cell-wall synthesis and cell-division processes. Importantly, due to its presence in the membrane, it is directly targetable with an antibody, thereby sensitizing the bacterium to an antibiotic concentration it is normally not sensitive to.

**The *ami*-operon encodes an antibiotic importer, and inhibition triggers tolerance.** The example above illustrates how negative fitness indicates increased antibiotic sensitivity reflected

by reduced relative growth, which can guide the development of (re)sensitizing approaches. In contrast, the occurrences of increased fitness in the dataset indicate that a large number of options exist that could lead to reduced antibiotic sensitivity (Fig. 3). With increased fitness to 3 out of 4 antibiotic classes, the *ami*-operon is among genes with the greatest number of positive fitness effects. The operon forms a tight cluster in the interaction network (Figs. 3 and 5a) and it is annotated as an oligopeptide transporter with no clear function. Two separate knockouts for SP_1888 (*amiE*) and SP_1890 (*amiC*) confirm that increased fitness results in decreased drug sensitivity in the form of increased relative growth in the presence of ciprofloxacin, vancomycin and gentamicin, and increased sensitivity (i.e., decreased relative growth) to Synercid (Fig. 5b). There is limited evidence that the *ami*-transporter may have (some) affinity for at least two different peptides (P1 and P2)[51–53]. These have been theorized to possibly function as signaling molecules and

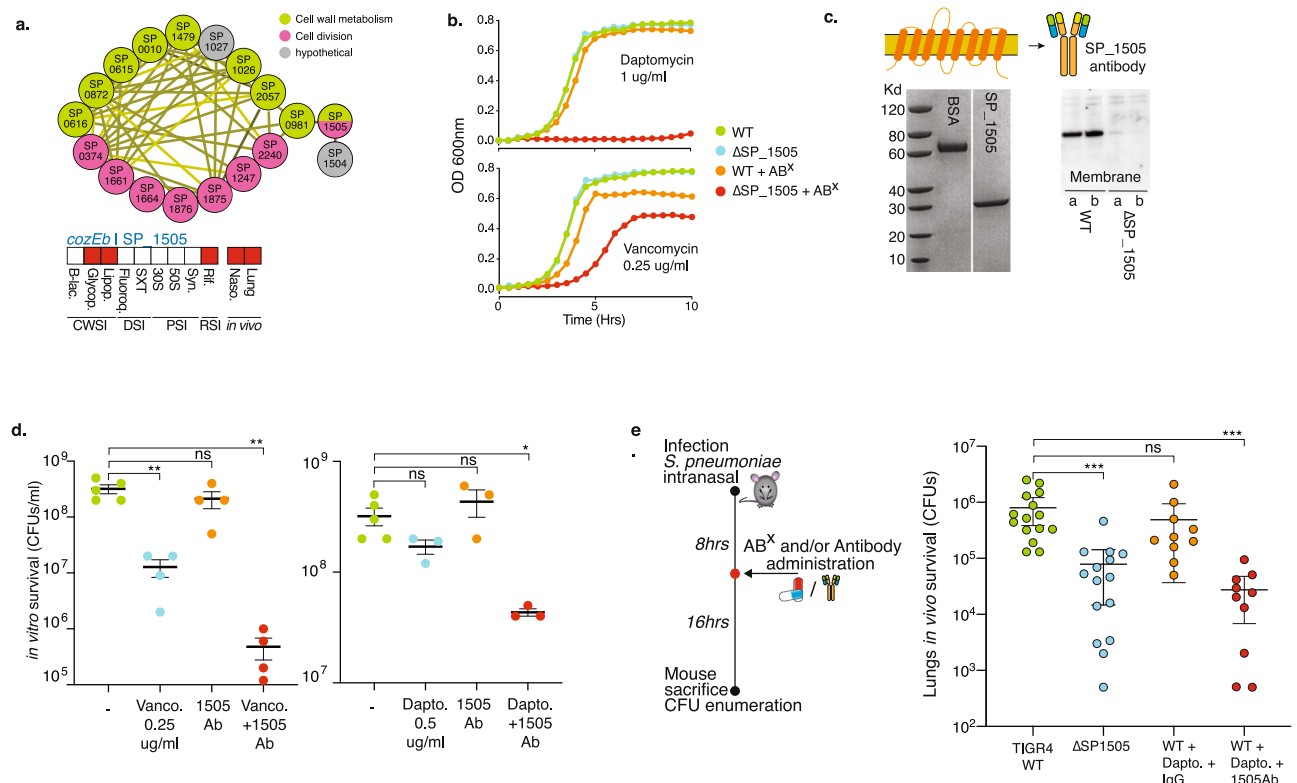

**Fig. 4 CozEb an integral membrane protein increases antibiotic sensitivity and can be targeted with an antibody. a** *cozEb*/SP_1505 is tightly clustered with cell division and cell wall metabolism genes, it is predicted to increase sensitivity to glycopeptides and the lipopeptide daptomycin, and has a decreased fitness in the mouse lung and nasopharynx. **b** Reduced relative growth of Δ*cozEb* validates its increased sensitivity to daptomycin and vancomycin. **c** CozEb has 8 transmembrane domains, which generates a ~30 Kd product (BSA is shown as a control). The cloned protein was used to raise antibodies, which proofed to be specific for a product in the WT membrane, but does not bind anything in Δ*cozEb*, indicating the antibodies are specific for the membrane protein CozEb. **d** Incubation of WT for 2 h with vancomycin (Vanco) or daptomycin (Dapto) and in the presence of CozEb antibody, slightly but significantly decreases bacterial survival. Mean values ± SEM are shown from n≥3 independent experiments. **e** An in vivo lung infection with WT or Δ*cozEb* confirms the mutant is less fit in vivo. Challenging the WT with daptomycin and IgG does not affect bacterial survival. In contrast, challenging with daptomycin and CozEb-specific antibodies, significantly reduces the recovered CFUs 24 h post infection. Mean values ± SEM are shown from $n \geq 10$ mice/experiment. Significance is measured through a one-way ANOVA with Dunnett correction for multiple testing: $^*p = 0.03$, $^{**}p = 0.001$, $^{***}p < 0.001$. Source data are provided as a Source Data file.

under certain circumstances may be generated by the bacterium itself[51–53]. Both peptides were synthesized and while neither peptide affects growth of the WT or knockout mutants in the absence of antibiotics (Supplementary Fig. 4), the WT grows slightly better in the presence of gentamicin and peptide P2, but not P1 (Fig. 5b). This shows that some peptides may, at least partially, inhibit or occupy the *ami*-transporter, and thereby trigger decreased antibiotic sensitivity, in a similar manner as a knockout does. Besides peptides, the *ami*-transporter may be (non-selectively) transporting antibiotics into the cell, which could explain its effect on antibiotic sensitivity. To explore this, bacteria were exposed to ciprofloxacin or kanamycin and the internalized antibiotic concentration was determined through mass spectrometry for WT and both *ami* knockout mutants. In both mutants the amount of internalized ciprofloxacin was significantly lower (~1.7× in Δ*amiE*, and ~2.3× in Δ*amiC*), while the kanamycin concentration was found to be significantly lower in Δ*amiC* (~2×; Fig. 5c). This shows that a functional *ami*-transporter increases the concentration of fluoroquinolones and 30S PSIs, suggestively by transporting them into the cell, and thereby, due to a higher internal concentration, enhancing the antibiotic's inhibitory effects on growth. There are multiple examples that transporters can contribute to tolerance[54,55], which we recently showed is also the case for the *ade* transporter in *Acinetobacter baumannii*, which contributes to fluoroquinolone tolerance[7]. However, those examples are mostly based on efflux pumps that actively decrease the antibiotic concentration in the cell through upregulation of such pumps. In contrast, with respect to the *ami*-operon it would be the reverse, i.e., inhibition instead of upregulation would lead to tolerance. To explore this possible effect on tolerance, the WT and Δ*amiE* were exposed to either 10xMIC of gentamicin or vancomycin over a period of 24 h. Approximately 1% of the WT population survives 4 h exposure to gentamicin, while none of the population survives exposure past 8 h. The Δ*amiE* population displays a slower decline in survival with 1% of the population surviving the first 8 h (tolerant cells)[25]. At ~10 h the decline ceases and the remaining population (~0.01%) survives at least up to 24 h, which is representative of a persister fraction[25]. In contrast, the WT and *amiE* mutant populations decline at similar rates when exposed to vancomycin, showing that inhibition of the *ami*-transporter can lead to tolerance and persistence in an antibiotic-specific manner while MICs of gentamycin and vancomycin for WT and Δ*amiE* are similar (Supplementary Data 1). Importantly, these data show that increased fitness indeed leads to decreased ABX sensitivity, which can translate into at least two phenotypes: increased relative growth and increased survival (i.e., tolerance).

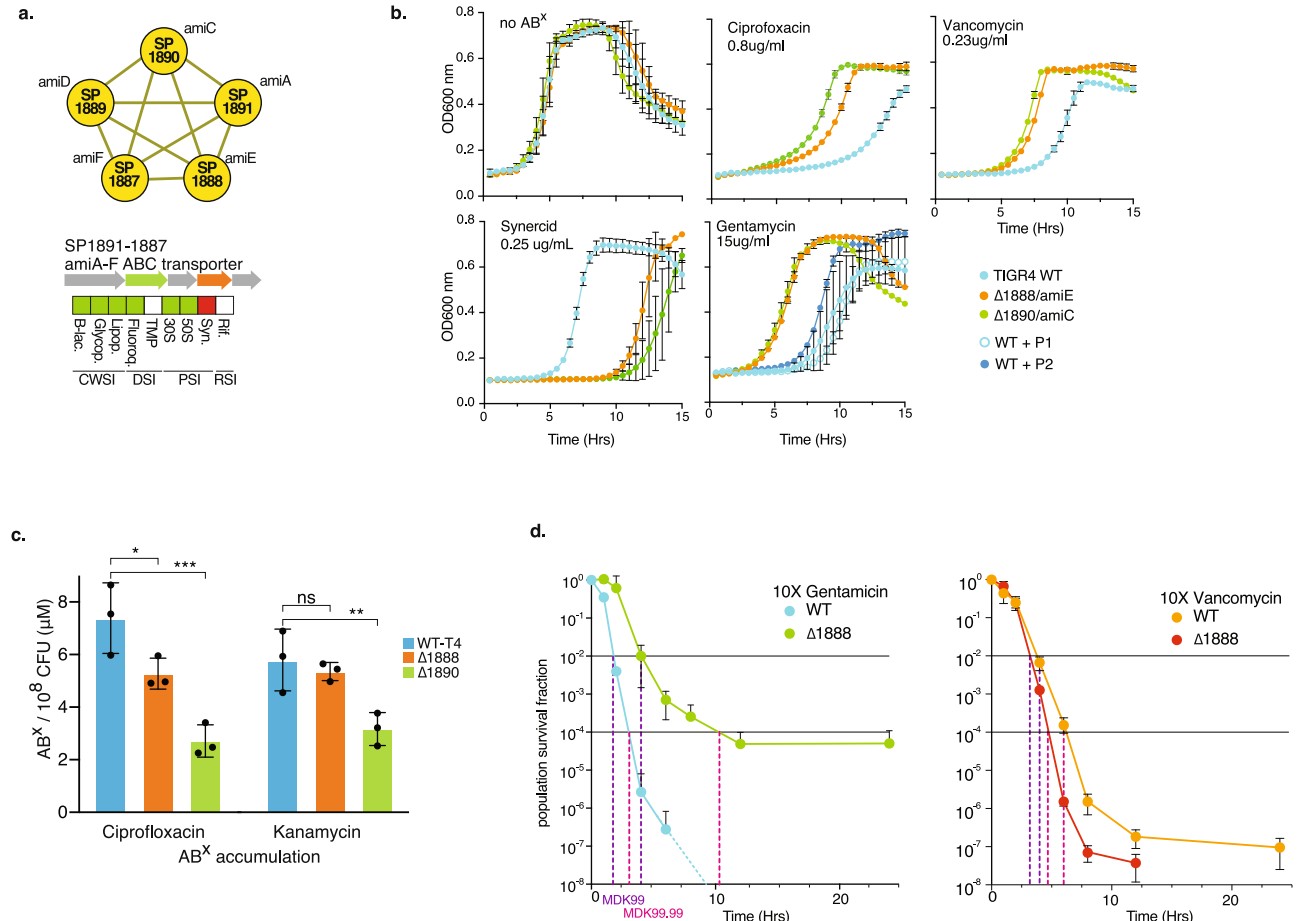

**Fig. 5 Modulation of the *ami*-transporter decreases sensitivity to many antibiotics. a** The *ami*-operon forms a tight cluster, and upon knockout is predicted to decrease sensitivity to most antibiotics, and increase sensitivity to Synercid. **b** Growth curves of individual knockout mutants of *amiE* and *amiC* validate changes in antibiotic sensitivity; i.e., they show that positive fitness translates into decreased ABX sensitivity and increased relative growth, while negative fitness translates into increased ABX sensitivity and decreased relative growth. Additionally, growth curves suggest the transporter phenotypically responds to peptide P2. Mean values ± SEM are shown from $n \geq 3$ independent experiments. **c** Intracellular antibiotic accumulation analysis shows that the WT strain with an intact transporter reaches a higher intracellular antibiotic concentration, suggesting the transporter is involved in importing antibiotics, explaining why a knockout or occupation with a peptide such as P2, can lead to decreased antibiotic sensitivity. Mean values ± SEM are shown from $n \geq 3$ independent experiments. **d** Besides that modulation of the transporter leads to positive fitness, which translates into decreased ABX sensitivity and increased relative growth in the presence of gentamicin or vancomycin, it also leads to increased survival (i.e., tolerance) to gentamicin, but not vancomycin. Mean values ± SEM are shown from $n = 4$ independent experiments. Significance is measured through a one-way ANOVA with Dunnett correction for multiple testing: *$p = 0.05$, **$p = 0.01$, ***$p = 0.001$. Source data are provided as a Source Data file.

**Purine metabolism, (p)ppGpp and ATP production are tightly linked to altered ABX susceptibility and tolerance.** Among the 21 functional groups, purine metabolism has some of the largest number of positive fitness effects, mostly with β-lactams and glycopeptides (Figs. 3a and 6a). Moreover, two regulators (SP_1821/1979) associated with this pathway decrease sensitivity to β-lactams and/or glycopeptides and two 'neighboring' genes with unknown function have either the same (SP_0830), or the opposite effect (SP_1446) on antibiotic sensitivity as their defined neighbor, suggesting they may be involved in the same process as their neighbor (Fig. 6a). Furthermore, the global interaction network positively links an ABC transporter (SP_0845-0848, Figs. 2c, 6a) with multiple genes in this pathway due to their similar profiles. This operon is annotated as a putative deoxyribose transporter, and to verify whether an interaction exists with purine metabolism, single and double knockouts were created between SP_0846 (the transporter's ATP binding protein) and SP_0829/*deoB*. Their profiles suggest they do not affect growth in the absence of ABXs and have increased sensitivity to Synercid, which was confirmed with individual growth curves (Fig. 6b).

However, when both knockouts are in the same background, their increased sensitivity to Synercid is masked. Thus, as indicated by the network, these results show that the ABC transporter indeed has a genetic interaction with purine metabolism/salvage, but plays an unknown role. Importantly, this confirms that the global network includes valuable interactions that can be explored to uncover functional relationships.

Furthermore, within purine metabolism the alarmone (p)ppGpp is synthesized from GTP and/or GDP. Like other bacterial species, *S. pneumoniae* likely responds to (some) ABXs via induction of the stringent response pathway[56], in which *relA* (SP_1645) is the key player with both synthetase and hydrolase activity[57]. Additionally, SP_1097 is annotated as a GTP diphosphokinase and may be involved in the synthesis of pppGpp from GTP (Fig. 6a). Our data suggests, and we confirmed for the β-lactam cefepime (Fig. 6c), that when synthesis of the alarmone is inhibited by deletion of *relA*, similar to many other interactions in purine metabolism, this leads to reduced β-lactam and glycopeptide sensitivity manifested by increased relative growth (Fig. 6c). Moreover, while SP_1097, as predicted, does not

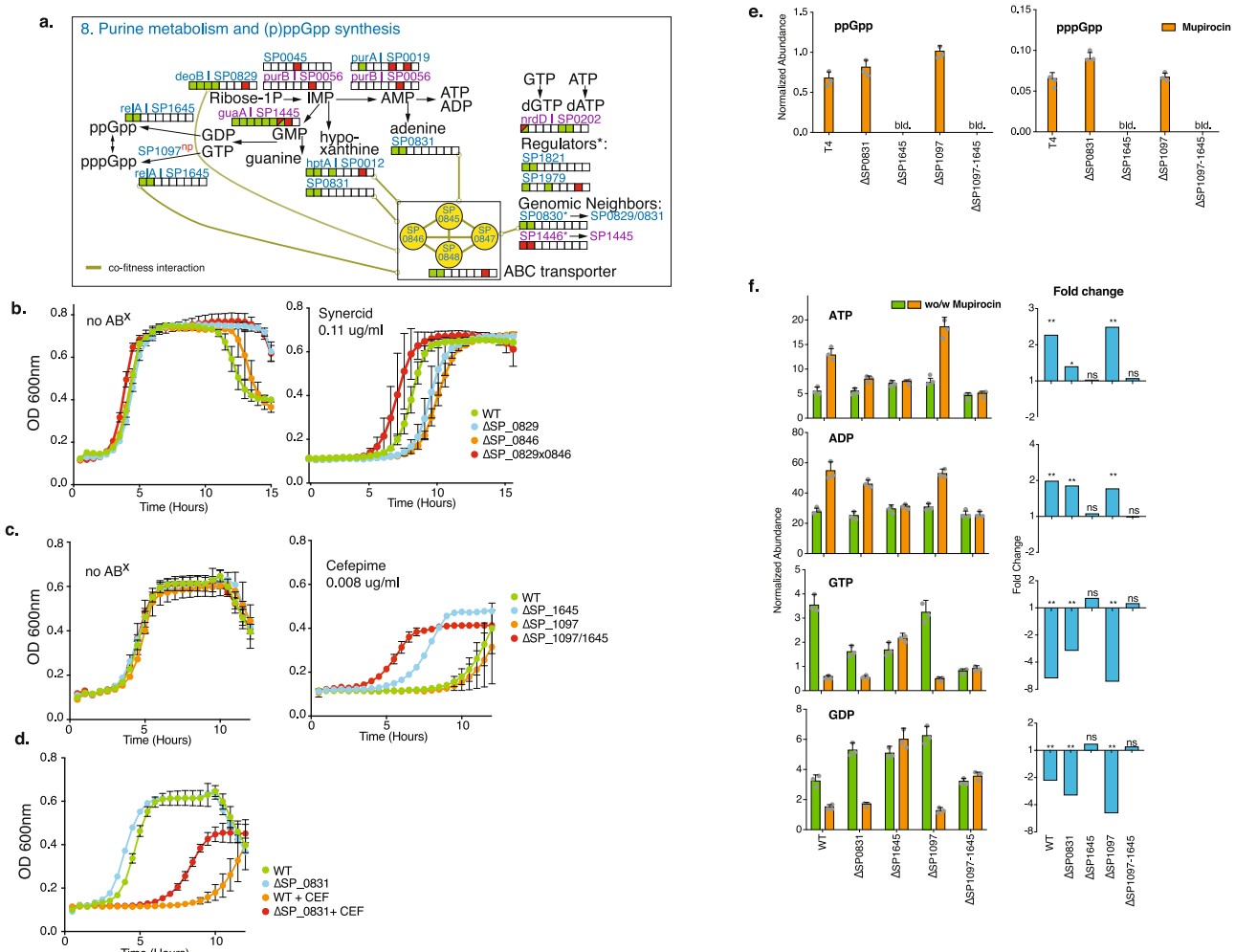

**Fig. 6 Modulation of purine metabolism affects (p)ppGpp and ATP synthesis and is linked to changes in ABX sensitivity. a** Key steps in purine metabolism with the same color coding as in Fig. 3. SP_1097 is listed as well, for which we found no change in ABX sensitivity, which is denoted with 'np' for no phenotype. The putative deoxyribose transporter (SP_0845-0848), a high-connectivity cluster in Fig. 2, is also shown. **b** Single knockouts for deoB/ SP_0829 and SP_0846, as well as a double knockout show that mutants and WT grow equally well in the absence of antibiotics. In the presence of Synercid, as predicted and indicated by their ABX sensitivity bar, the single knockouts display a higher sensitivity to the drug then the WT. The double mutant suppresses the increased Synercid sensitivity phenotype of the single mutants, indicating that the positive interaction that is found in the co-fitness network leads to a positive genetic interaction between these genes. **c** Single and double knockouts of SP_1097 and SP_1645/relA grow just as well as WT in the absence of antibiotics. As predicted SP_1097 is equally sensitive to cefepime as the WT, while ΔrelA has decreased sensitivity as indicated by its ABX sensitivity bar in **a**. Additionally, the double knockout has decreased sensitivity to cefepime, indicating the dominant phenotype of ΔrelA. **d** The phenotype of ΔSP_0831 was validated in growth as well, showing no change in growth in the absence of ABX, and decreased sensitivity (i.e., increased relative growth) in the presence of cefepime (FEP). **e** The alarmone (p)ppGpp is below the limit of detection (b.l.d.) in the absence of stress, upon induction with mupirocin it is synthesized in equal amounts in WT, ΔSP_0831 and ΔSP_1097, while it cannot be synthesized when relA is absent. **f** Synthesis of di- and trinucleotides is significantly affected in the different mutants upon mupirocin exposure. Mean values ± SEM are shown from n≥3 independent experiments. Significance is measured through a paired t-test with an FDR adjusted p value for multiple comparisons: *p < 0.05, **p < 0.01, ***p < 0.001, ns not significant. Source data are provided as a Source Data file.

change ABX sensitivity (Supplementary Data 2, Fig. 6), a double knockout of *relA*-SP_1097 seems to further decrease sensitivity to cefepime by further increasing relative growth (Figs. 6c and 7a). Additionally, besides a change in growth, the single *relA* and double knockout (Δ*relA*-SP_1097), also increases tolerance to cefepime by ~1000-fold at 24 h (Fig. 7b), without changing the MIC (Supplementary Data 1). To understand how *relA* and SP_1097 affect purine metabolism, we used LC/MS to measure (p)ppGpp, ADP, ATP, GDP, and GTP. Additionally, we included SP_0831 a purine nucleoside phosphorylase involved in nucleotide salvage, which has the same ABX profile as Δ*relA* (Fig. 6a, d), but should not directly affect (p)ppGpp synthesis. While (p) ppGpp is below the limit of detection during normal growth in

any of the strains, as expected Δ*relA* and the double mutant Δ*relA*-SP_1097 are unable to synthesize the alarmone when exposed to mupirocin, a strong activator of the stringent response (Fig. 6e, Supplementary Data 9). In contrast, WT, ΔSP_0831, and ΔSP_1097 synthesize (p)ppGpp upon mupirocin exposure to a similar extent (Fig. 6e). Concerning the di- and trinucleotides in the pathway, upon mupirocin exposure GTP and GDP are significantly reduced in WT, ΔSP_0831, and ΔSP_1097, likely because they are used for (p)ppGpp synthesis (Fig. 6f, Supplementary Data 9). In contrast, while ATP and ADP again remain constant for the Δ*relA* mutants, ATP and ADP synthesis are significantly increased upon mupirocin exposure, especially for WT and ΔSP_1097. This suggests that during activation of the

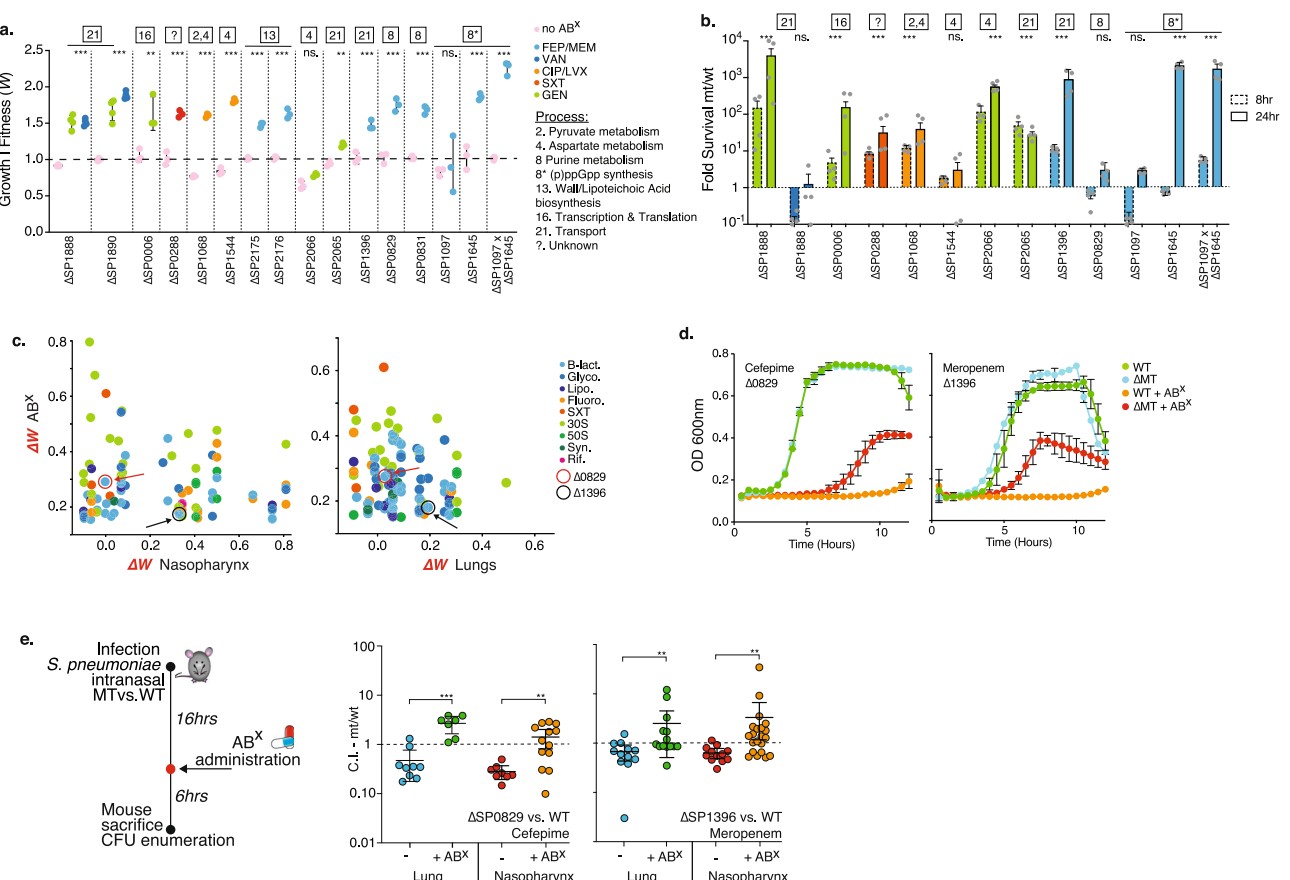

**Fig. 7 Decreased antibiotic sensitivity and tolerance can be achieved by modulation of a wide variety of processes. a** Relative growth rates (i.e., fitness) of 16 knockout mutants involved in 7 processes measured in the presence of 7 antibiotics, validate that decreased ABX sensitivity (i.e., increased relative growth) can be achieved by modulating a wide variety of processes. Mean values ± SEM are shown from n≥3 independent experiments. **b** Significantly increased survival during exposure to 5–10xMIC of an ABX over a 24 h period is observed for 9 out of 12 knockouts. Significance is measured with an ANOVA with Dunnett correction for multiple comparisons: **$p < 0.01$, ***$p < 0.001$. Mean values ± SEM are shown from $n = 4$ independent experiments. **c** Tn-Seq data with a positive fitness in the presence of at least one antibiotic (y-axis) is plotted against in vivo Tn-Seq data (x-axis). Note that only in vivo data is shown that is predicted to have no more than a small fitness defect, no fitness defect or an increased predicted in vivo fitness, either during nasopharynx colonization or lung infection. Circled and indicated with arrows are SP_0829 in red and SP_1396 in black. **d** In vitro growth curves validate decreased sensitivity (i.e., increased relative growth) to cefepime (SP_0829) and meropenem (SP_1396). Mean values ± SEM are shown from $n = 3$ independent experiments. **e** Mice were challenged with WT and MT in a 1:1 ratio of which half received ABX 16 h post infection (p.i.), and all were sacrificed 24 h p.i. Displayed are the MT's competitive index (CI) in the nasopharynx and lung, and in the presence and absence of cefepime (SP_0829) or meropenem (SP_1396). In all instances, the addition of ABX significantly increases the CI of the mutant. Significance is measured with a Mann–Whitney test **$p < 0.01$, ***$p < 0.001$. Mean values ± SEM are shown from $n ≥ 7$ mice/experiment. Source data are provided as a Source Data file.

stringent response, synthesis from IMP is directed toward AMP, and not necessarily GMP, at least not enough to replenish GTP and GDP. Additionally, upon mupirocin exposure, ATP only minimally increases for ΔSP_0831, while it increases over twofold for WT and ΔSP_1097 (Fig. 6f). It has been shown for bacteria including *Escherichia coli* and *Staphylococcus aureus* that a decreased ATP concentration can decrease sensitivity to ABXs such as ciprofloxacin[58]. Additionally, in *S. aureus* (p)ppGpp overexpression has been associated with decreased sensitivity to linezolid[59]. Our data suggest that (p)ppGpp and ATP synthesis may be intrinsically linked, i.e., at least in *S. pneumoniae* the inability to produce the alarmone also results in lowered ATP synthesis, which is associated with a lowered ABX sensitivity to β-lactams and glycopeptides. However, ΔSP_0831 shows that even if (p)ppGpp can be synthesized, modulation of purine metabolism, for instance through the salvage pathway, can result in decreased ATP synthesis, and can lead to lowered ABX sensitivity (i.e., increased relative growth). Importantly, in many bacterial species, alarmone production is generally assumed to be triggered

in response to different types of stress and has been shown to affect a large variety of processes including nucleotide synthesis, lipid metabolism, and translation. (p)ppGpp is thereby a ubiquitous stress-signaling molecule that enables bacteria to generate a response that is geared toward overcoming the encountered stress. However, contradictory results between species indicate a possible non-uniformity across bacteria, leaving much to be learned about how the alarmone and the processes it can control fit into the entire organismal (response) network[56]. Our data suggest that the inability (i.e., due to mutations) to generate the alarmone in *S. pneumoniae* in response to β-lactams and glycopeptides is linked to reduced ATP, which under specific circumstances may be an optimal response, as it results in decreased ABX sensitivity translating into increased relative growth and tolerance, and thereby a higher probability to survive the insult (Figs. 6c and 7a, b).

**There are a multitude of predictable pathways that lead to tolerance in vivo in an antibiotic dependent manner.** For instance,

in the glycolysisTo further confirm that antibiotic sensitivity can be decreased by inhibiting a variety of processes, knockouts (KOs) were generated for fourteen mutants from 8 different processes. Moreover, an additional goal was to determine what increased fitness (i.e., decreased ABX sensitivity) would look like phenotypically, and thus whether it would translate into increased relative growth and/or tolerance. Of the 14 mutants with a Tn-Seq predicted increased fitness, 13 display an increased ability to grow in the presence of an ABX compared to the WT. Moreover, eight mutants, which inhibit several different processes including different metabolic pathways, transport, and transcription and translation, displayed tolerance, while retaining a similar MIC, and thereby have an increased ability to survive high-level exposure to an ABX (5–10xMIC) for at least 24 h (Fig. 7a, b, Supplementary Data 1,8). Note that we validated 49 single KO genotype × phenotype associations in this study, with an equal distribution across the entire spectrum of ABX sensitivity (Fig. 1e, Supplementary Data 8). This highlights that our approach uncovered a detailed genome-wide ABX sensitivity atlas composed of a multitude of genes, pathways and processes that when modulated can increase and/or decrease ABX sensitivity. The validation experiments highlight that the resulting fitness accurately predicts the relative growth rate of a mutant, which we have previously shown for hundreds of other negative fitness phenotypes[6,14–16,18,39–42,44,60–62]. Moreover, it turns out that in the majority of cases, increased fitness not only results in increased relative growth in the presence of an antibiotic, but also tolerance. Thereby, the part of the atlas that depicts decreased ABX sensitivity (i.e., increased fitness) includes a genome-wide 'tolerome', composed of a wide variety of pathways and processes that when modulated trigger tolerance in vitro in an ABX dependent manner.

Obviously, the selection regime in vivo is far more complex and stricter than in a test tube, which raises the question whether many of the options that decrease ABX sensitivity in vitro, including those that increase tolerance, would be available in vivo as well. To explore this, all the Tn-Seq data with a positive fitness in the presence of at least one antibiotic was combined with in vivo Tn-Seq data and filtered for those genes with no or only a small fitness defect predicted in vivo during nasopharynx colonization or lung infection (Fig. 7c, Supplementary Data 2). Two genes were selected that we had confirmed for decreased ABX sensitivity in vitro: (1) SP_0829/deoB synthesizes Ribose-1P and is involved in purine metabolism (Fig. 6a). ΔdeoB has no effect on in vitro growth in the absence of ABX (Fig. 7a, d), as predicted it grows better in the presence of cefepime (Fig. 7a, d), but it does not affect survival/tolerance (Fig. 7b); (2) SP_1396/pstA is the ATP binding protein of a phosphate ABC transporter (Supplementary Fig. 3). ΔpstA has no effect on in vitro growth (Fig. 7a, d), it has a higher relative growth rate in the presence of meropenem (Fig. 7a, d), and it also increases survival/tolerance (Fig. 7b). Both mutants were mixed with WT in a 1:1 ratio and used in an in vivo mouse infection competition model as we have done previously[18]. Of the infected mice, half were administered antibiotics at 16 h post infection, and were sacrificed 6 h later to determine the strain's competitive index (CI) (Fig. 7e). Importantly, while both mutants may have a slight disadvantage compared to the WT when colonizing the lung or nasopharynx, their CI increases significantly in the presence of ABXs, leading to increased survival compared to the WT (Fig. 7e, Supplementary Data 10). Combining antibiotic- with in vivo Tn-Seq highlights the ability to predict the existence of a wide array of possible alterations of specific genes, pathways and processes that can have a beneficial effect in vivo in the presence of antibiotics. Such changes could thereby contribute to escape from antibiotic

pressure and even create a path toward the emergence of antibiotic resistance.

There is likely significant overlap in the selective pressures a bacterial pathogen would experience in a mouse infection model compared to the human host. This raises the possibility that those gene disruptions that are predicted by Tn-Seq to lead to decreased antibiotic sensitivity and that simultaneously have no more than a minimal defect in vivo, could also have an advantage in the human host in the presence of ABXs and thereby contribute to ABX escape and/or the emergence of resistance. A premature stop codon most closely reflects the effect a transposon insertion has on a gene; i.e., it disables a gene. We thereby hypothesized that stop codons in certain gene sets predicted by Tn-Seq could be enriched for in antibiotic-resistant clinical isolates. To test this hypothesis 4 gene sets were compiled consisting of those that upon disruption: (1) decrease antibiotic sensitivity in at least 1 antibiotic and have no strong defect in vivo; (2) decrease antibiotic sensitivity in at least 1 antibiotic and have a defect in vivo; (3) have little to no effect on antibiotic sensitivity and in vivo; (4) have no effect or increase antibiotic sensitivity and have a defect in vivo (Fig. 8a, b; Supplementary Fig. 5). Thousands of strains were selected from the PATRIC[63,64] database that could be split into a group of co-trimoxazole (SXT) resistant and a group of β-lactam resistant strains, and each group was matched with an equal number of sensitive strains from the database. In all strains in the SXT and β-lactam groups, irrespective of resistant or sensitive status, the number of stop codons in gene sets 1 and 3 are highest, which reflects the Tn-Seq predicted in vivo effects, i.e., while gene sets 1 and 3 contain mostly genes with potentially neutral effects, gene sets 2 and 4 contain many genes that are suggested to have a defect in vivo when disabled (e.g. with a stop codon) (Fig. 8c). Moreover, SXT resistant isolates in gene set 1 more often contain a stop codon compared to sensitive strains, and in β-lactam resistant isolates this is true for gene sets 1–3 (Fig. 8d). While these are not ideal comparisons, for instance the entire ABX profile is not clear for many strains, different changes than premature stops could have ABX/in vivo modulating effects, strains could have experienced different ABX and/or in vivo selective pressures, and genetic changes can be strain-background dependent, it shows that genetic changes that can affect ABX and/or in vivo sensitivity, which are predictable with Tn-Seq, readily occur in clinical samples. This in turn underscores that ongoing infections may consist of variants that enable different paths to adjusting to, or overcoming a challenging host/ABX environment.

## Discussion

The emergence and increase in antibiotic resistance among most bacterial pathogens is a continuously developing problem with several important drivers, which include: (1) a lagging development of new drugs and treatment strategies; (2) a lack of (rapid) diagnostics and prognostics; and (3) an incomplete understanding of how antibiotic resistance develops. Moreover, these drivers are inherently connected making it a complex problem to solve. First, the ability of bacteria to evolve resistance elicits an arms-race that requires the development of new drugs and treatment strategies to keep the balance of infection-control tipped in our favor. Thus, while developing new drugs would keep the arms-race in place, the ability to slow or prevent the emergence of resistance could resolve the status quo. Furthermore, even though it is critical to understand how and under which circumstances resistance evolves, the applicability of this knowledge depends on the availability of diagnostics that could inform on the emergence of resistance (precursors) and thereby guide

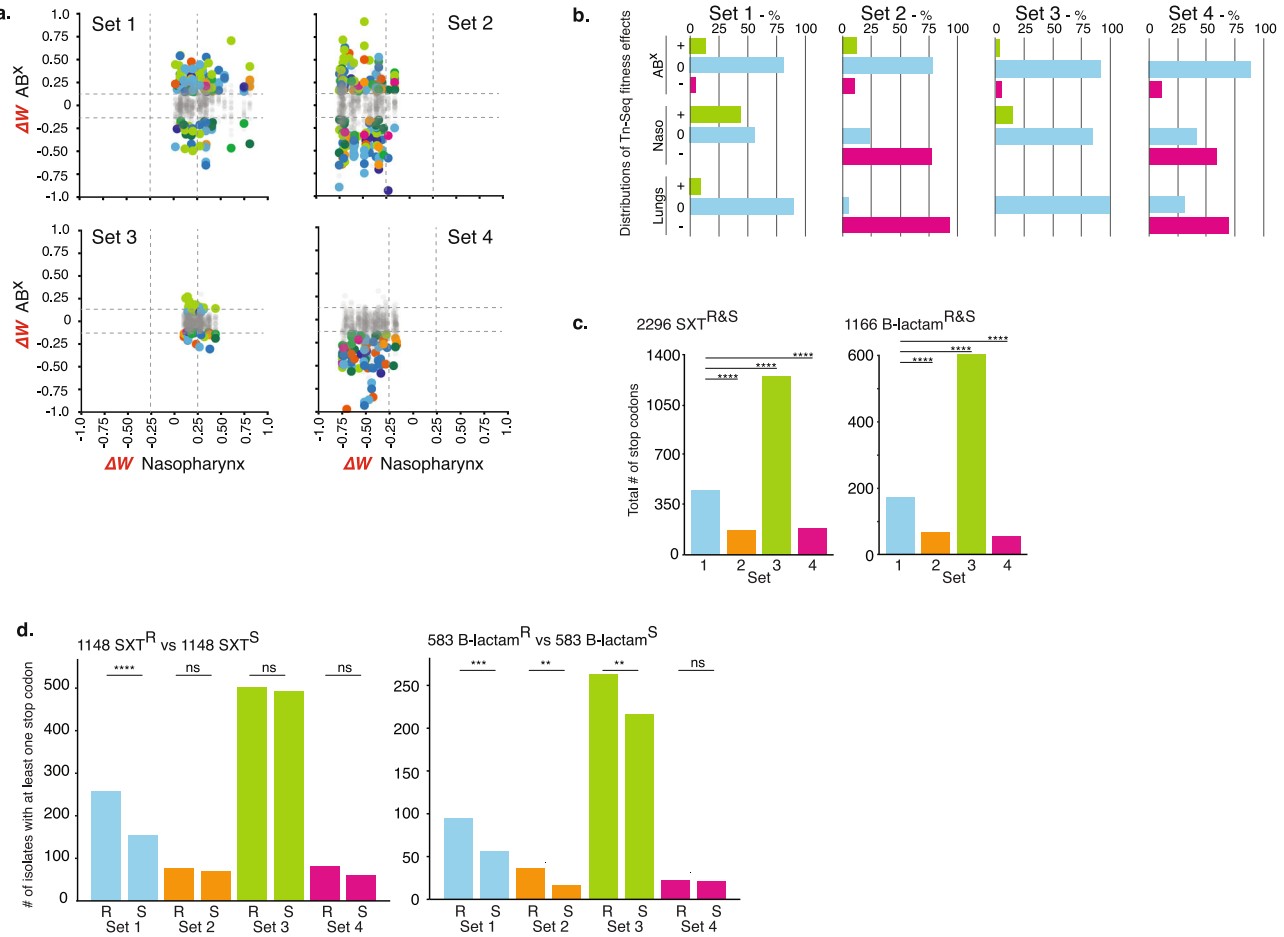

**Fig. 8 Stop codons are enriched in clinical samples in Tn-Seq predicted tolerome genes. a** Based on in vivo and ABX Tn-Seq data, four gene sets consisting of 34 genes each were compiled with specific fitness profiles in the presence of antibiotics and in vivo. Shown are the in vivo effects for nasopharynx, while lung data are depicted in Supplementary Fig. 5. $\Delta W$ represents the fitness difference of a gene in a specific condition (e.g., an antibiotic, in vivo) minus its fitness in vitro in rich medium. Dashed lines indicate significance cut-offs, grayed-out dots indicate genes with no significant change in fitness in the presence of antibiotics, colors represent antibiotics and are the same as in Fig. 1. **b** Detailed distributions for each gene set highlight whether effects in the presence of antibiotics, in the nasopharynx and lungs increase (+), do not affect (0) or decrease (−) relative fitness. Gene set rationales are described in the text. **c** The total number of stop codons in each gene set for 2296 co-trimoxazole and 1166 β-lactam resistant and sensitive strains. **d** The number of sensitive and resistant strains with at least one stop codon in a gene in each gene set. Significance is measured through a Fisher's exact test: **$p < 0.01$, ***$p < 0.001$, ****$p < 0.0001$. Source data are provided as a Source Data file.

and enable timely, tailored, and targeted treatments. To progress toward a comprehensive understanding of how an infection is developing in the absence or presence of treatment, and how to decide what to do next, we believe that a detailed genetic understanding of how a bacterium deals with and overcomes stress, as well as its genetic potential to achieve this, are key aspects. In this study, we contribute to reaching such an understanding by building and exploring a detailed atlas of ABX sensitivities, which highlights how modulation of specific genes, pathways and processes does not only result (as expected) in increased ABX sensitivity, but surprisingly often in decreased ABX sensitivity. We show that such an atlas can be used to identify leads for gene function, to uncover the genome's underlying architecture and genetic relationships among genes, for the identification of new drug targets, and the development of new proof-of-principle antimicrobial (ABX sensitizing) strategies. Most importantly, these data identify genome-wide genetic changes that show how modulation of genes, pathways, and processes can lead to reduced antibiotic sensitivity (i.e., increased relative growth and tolerance), not only in vitro, but also in vivo. Moreover, we show that mutations that have the potential to trigger the same phenotypes readily occur in patients.

While the primary processes targeted by ABXs are mostly known, this work contributes to the increasing notion that downstream processes, not directly related to the target and which include metabolism, can significantly contribute to antibiotic efficacy[14,15,18,65–68]. In E. coli it has been shown that inhibition of specific steps in purine metabolism can lead to decreased sensitivity to ampicillin and ciprofloxacin, and increased sensitivity to gentamycin[69]. Our data explores a wider set of alterations in purine metabolism in S. pneumoniae, which also leads to a wider distribution of changes. However, the overlap with E. coli includes decreased sensitivity to β-lactams (an ABX class that includes ampicillin) and we show that inhibition of some reactions in purine metabolism can also lead to decreased sensitivity to fluoroquinolones (an ABX class that includes ciprofloxacin). Additionally, it has been shown for E. coli that antibiotic sensitivity to ampicillin and ciprofloxacin can be increased, at least over 4 h, by supplementing with adenine, but not guanine, which is possibly linked to a reduced ATP demand and synthesis[69]. Our results show that limiting the ability to synthesize adenine, but also guanine, leads to significantly lowered ABX sensitivity to β-lactams and glycopeptides. Moreover, we show that limiting the ability to synthesize adenine, as well as

(p)ppGpp, leads to lowered ATP synthesis. The association between decreased ATP synthesis/availability and decreased sensitivity and/or tolerance to antibiotics including β-lactams, glycopeptides and fluoroquinolones has now been shown for a variety of Gram-positive and -negative organisms[35,58,70,71]. While this suggests that low ATP demand/synthesis/availability may be at the root of a general mechanism that leads to decreased sensitivity to some antibiotics, our genome-wide atlas shows that decreased ABX sensitivity and tolerance to β-lactams, and glycopeptides can be triggered by alterations to pathways and processes other than those directly related to ATP synthesis, which include parts of glycolysis, pyruvate, ascorbate, glucose and purine metabolism, protein turnover and c-di-AMP synthesis (Supplementary Figs. 2 and 3). While, it is possible that many of these alterations do affect ATP demand, availability and/or synthesis, it is likely that they trigger a much more complex and varied set of changes. Importantly, the full extent of signals and (genetic) alterations that can lead to decreased ABX sensitivity, including tolerance, remain mechanistically poorly understood[25,34]. This also means that it remains unclear whether there are common denominators or universal rules that are applicable across strains and species[7,25,34,35,58,70,71]. Different (computational) approaches are being explored to build such a comprehensive understanding[6,15,17,69,72,73], however, unequivocally more genome-wide data from more species remain needed. Thereby, the genome-wide insights we present here are helping build a rationale to measure and model this complexity. One goal of such models would be to obtain a detailed understanding and ability to predict how alterations to specific processes affect responses to ABXs and thereby drive changes in sensitivity[15,17,25,34,69]. Lastly, these detailed data on reduced antibiotic sensitivities also suggest that more potential routes to ABX escape, and eventually resistance, may exist than assumed. We believe these data are thereby both an argument and potential starting point for a platform to predict clinically relevant mutations and determinants of antibiotic resistance and/or tolerance. Consequently, they underscore the importance of understanding the genetics of variants with altered drug susceptibility, as their genetics makes them diagnostically identifiable and trackable, while their often-associated collateral sensitivities to other ABXs or drugs could make them targetable.

## Methods

The research presented within complies with all relevant ethical regulations and protocols as approved by the Boston College environmental health and safety board.

**Bacterial culturing, growth curves, and tolerance experiments**. Experiments were performed with *S. pneumoniae* strain TIGR4 (NCBI Reference Sequence: NC_003028.3). TIGR4 is a serotype 4 strain that was originally isolated from a patient from Norway with Invasive Pneumococcal Disease (IPD)[74,75]. All 'SP_' gene numbers in the tables and figures are according to the TIGR4 genome. Single gene knockouts were constructed by replacing the coding sequence with a chloramphenicol and/or spectinomycin resistance cassette as described previously[18,39,40]. *S. pneumoniae* was grown on sheep's blood agar plates or statically in THY, C + Y or semi-defined minimal media at pH 7.3, with 5 μL/mL Oxyrase (Oxyrase, Inc), at 37 °C in a 5% $CO_2$ atmosphere[15]. Where appropriate, cultures and blood plates contained 4 μg/mL chloramphenicol (Cm) and/or 200 μg/mL spectinomycin (Spec). Single strain growth assays were performed three times using 96-well plates by taking $OD_{600}$ measurements on a Tecan Infinite 200 PRO plate reader or BioSpa 8 (BioTek). Growth curves are fitted to an exponential growth equation to calculate their doubling time. WT doubling time is divided by a mutant's (MT) doubling time to represent each mutant's fitness (i.e., relative growth rate; $WT_{doubling\ time}/MT_{doubling\ time} = W_{mutant}$), making it directly comparable to Tn-Seq fitness[14,15,18,39–41,44,60,62,76,77]. Tolerance experiments were performed by exposing exponentially growing bacteria to 5–10xMIC of an antibiotic. Samples were taken at different time points over a 24 h period, washed with PBS, and plated on blood agar for enumeration. The number of surviving bacteria at different time points are divided by the starting population to determine the

surviving proportion at each time point. The proportion of surviving MT bacteria are divided by the proportion of surviving WT bacteria to determine the fold survival (MT/WT) at each time point as depicted in Fig. 7b, Supplementary Data 8.

**Tn-Seq experiments, fitness (*W*) and enrichment analyses**. Six independent transposon libraries, each containing ~10,000 insertion mutants, were constructed with transposon Magellan6 in WT-T4 as described previously[14,18,39,77]. Selection experiments were conducted in rich medium with glucose as a carbon source in the presence or absence of 20 different antibiotics at a concentration that slows growth by ~30–50% (Supplementary Data 1). Libraries are grown in ~10 mL medium (wo/w ABX), from a starting $OD_{600}$ of ~0.003 (~40,000 CFU/mL) up to an $OD_{600}$ of ~0.3–0.6 (~1.10[7] CFU/mL), representing ~7–8 generations. Sample preparation, Illumina sequencing and fitness calculations were done as described[14,18,39,44,60,77]. Sequencing analyses and fitness calculations are performed with analysis platform Aerobio v2.3[78]. In short, fitness of a single mutant ($W_i$) is calculated by comparing the fold expansion of the mutant to the fold expansion of the population and is determined by the following exponential growth equation we previously developed[18,39,60]:

$$W_i = \frac{\ln\left(N_i(t_2) \times d/N_i(t_1)\right)}{\ln\left(\left(1 - N_i(t_2)\right) \times d/\left(1 - N_i(t_1)\right)\right)} \quad (1)$$

in which $N_i(t_1)$ and $N_i(t_2)$ are the mutant frequency at the beginning and end of the experiment respectively and $d$ is the population expansion. All of the insertions in a specified region or gene are then used to calculate the average fitness and standard deviation of the gene knockout in question. The advantage of using this approach is that $W_i$ represents the actual growth rate per generation, which makes fitness independent of time and enables comparisons between conditions. To determine whether fitness effects are significantly different between conditions three requirements have to be fulfilled: (1) $W_i$ is calculated from at least three data points, (2) the difference in fitness between the presence and absence of antibiotic has to be larger than 15% (thus $W_i - W_j = <-0.15$ or $>0.15$), and (3) the difference in fitness has to be significantly different in a one sample *t*-test with Bonferroni correction for multiple testing[18,39,60]. Importantly, we have previously validated that a mutant's Tn-Seq fitness is indeed directly related to its relative growth rate, which means that a mutant with for instance a Tn-Seq fitness of 0.5 ($W_i = 0.5$) grows twice as slow as the WT. In this study, we show that fitness is also directly related to a mutant's relative growth rate, where negative fitness indicates increased ABX sensitivity and decreased relative growth, while positive fitness indicates decreased ABX sensitivity and increased relative growth. Moreover, while Tn-Seq selection is performed under ABX pressure during growth for 7–8 generations we find here that positive fitness in many cases also leads to increased survival, i.e., tolerance.

In multiple figures $\Delta W$ ($W_{ABX} - W_{noABX}$) is displayed, which means that each gene's antibiotic-specific fitness is statistically compared to baseline fitness without ABXs. $\Delta W$ thereby indicates a gene's antibiotic-specific fitness effect which can be categorized as: (1) Neutral, $\Delta W = 0$, a mutant's relative growth is similar in the absence and presence of an ABX; (2) Negative, $\Delta W < 0$, a mutant's fitness is significantly lower and thus grows relatively slower in the presence of an ABX; (3) Positive, $\Delta W > 0$, a mutant's fitness is significantly higher and thus grows relatively faster in the presence of an ABX. For instance, if a gene knockout's $\Delta W = 0.25$, it means that its relative growth rate is ~25% higher in the presence of an ABX than in the absence of an ABX.

To determine whether a particular process or pathway is specifically involved in responding to an antibiotic group, a hypergeometric test was performed to test for enrichment. The distribution of significant genes within each process was compared to the distribution of the pathways in the overall genome. A *p* value and Benjamini–Hochberg adjusted *p* value were calculated for each process and antibiotic group, where an adjusted *p* value below 5% is considered to identify statistical enrichment.

**Determination of minimum inhibitory concentration (MIC)**. MICs were determined as previously described in ref. [17]. In short, ~1 × 10[5] CFU of mid-exponential bacteria are cultured in 200 μL in 96-well plates in fresh medium containing a single antibiotic at the following concentration gradients and increments: Ciprofloxacin gradient 0.4–1.8 μg/mL with 0.1 μg/mL increments; Cefepime 0.0175-0.03 μg/mL with 0.0025 μg/mL increments; Gentamicin 17–39.5 μg/mL with 2.5 μg/mL increments; Meropenem 0.006-0.02 μg/mL with 0.002 μg/mL increments; co-trimoxazole 4.5–10 μg/mL with 0.5 μg/mL increments; vancomycin 0.12–0.32 μg/mL with 0.04 μg/mL increments. MICs were determined in triplicate and monitored on a Tecan Infinite 200 PRO plate reader or BioSpa 8 (BioTek) at 37 °C for 12 h. MIC is determined as the lowest concentration that abolishes bacterial growth (Supplementary Data 1).

**Co-fitness network construction and SAFE analysis**. A gene x condition matrix was constructed to identify correlating fitness profiles and built a co-fitness network. The matrix is based on 20 antibiotic conditions from experiments performed here, supplemented with 17 conditions from van Opijnen and Camilli[18] (Supplementary Data 3). The additional conditions consist of sucrose, fructose, cellobiose,

raffinose, sialic acid, galactose, mannose, maltose, GlcNac, bipyridyl, transformation, hydrogen-peroxide, methyl-methane sulfonate, pH6, temperature, Norfloxacin. Genes with missing data were removed resulting in a 1519 gene × 37 condition matrix (Supplementary Data 3). Genes and conditions were correlated using a Pearson's correlation coefficient and a Spearman's correlation coefficient. Resulting in two 1519 × 1519, gene vs gene matrices. A significance cut-off was applied and correlations ≥0.75 were retained and used as edges to build a co-fitness network consisting of 1519 genes and 2399 edges. An edge-weighted spring embedded layout was applied with Cytoscape[79], with the absolute correlation value as the edge weight. This results in a network with several major clusters and multiple genes unconnected to the main network. A stability test was performed to determine the robustness and quality of each edge in the network by building a correlation matrix from partial data. Thirty conditions were selected 100 times to build a correlation matrix and using the same cut-off criteria a co-fitness matrix was compiled. Every edge with a correlation value above the threshold was assigned a 1 and every edge below the cut-off 0. This resulted in 100 binary matrices which were then summated, resulting in every gene vs gene interaction being assigned a stability score with a value N out of 100. A SAFE (Spatial Analysis of Functional Enrichment) analysis[45,46] on the co-fitness network was performed with Cytoscape. A SAFE analysis is geared toward defining local neighborhoods for each node within a network and calculates an enrichment score for every functional attribute. It then highlights the areas that are the most enriched for that attribute. Attributes were assigned by merging KEGG[80] pathway annotation and available functional category annotations, which covers 94% of the genes within the network. The distance threshold was set to the 1st percentile of the map-weighted distance, the Jaccard similarity index was set to 0.5, and nodes in different landscapes were retained.

### CozEb (SP_1505) cloning and protein expression.
Cloning and expression of SP_1505 was undertaken commercially (Genscript). Codon-optimized SP_1505 was cloned into pET28a with a C-terminal His-tag. *E. coli* BL21 (DE3) was transformed with recombinant plasmid. A single colony was inoculated into LB medium containing kanamycin; cultures were incubated at 37 °C at 200 rpm and IPTG was introduced for induction. SDS-PAGE and western blot were used to monitor the expression. Protein was purified from 1 L batch culture in Terrific Broth. Cells were harvested by centrifugation, cell pellets were lysed by sonication, and supernatant after centrifugation was kept for future purifications. SP_1505 protein was obtained by three-step purification using Ni column, Superdex 200 column, and Q Sepharose. Fractions were pooled and dialyzed followed by 0.22 µm filter sterilization. Protein was initially analyzed by SDS-PAGE and Western blot by using standard protocols for molecular weight and purity measurements. The concentration was determined by BCA protein assay with BSA as a standard. Final protein product was stored in 50 mM Tris-HCl, 150 mM NaCl, 10% Glycerol, 0.2% DDM, pH 8.0 and stored at −80 °C. Lab confirmation of the expression construct was undertaken using a 1:3000 dilution of mouse-anti-His-antibody, sourced from LIFE TECHNOLOGIES (catalog #37-2900) followed by Goat Anti-Mouse IgG (H + L)-HRP Conjugate (Biorad, catalog# 1706516), used at 1:5000 dilution.

### CozEb (SP_1505) antibody generation, purification and quantification.
A single rabbit was vaccinated by a commercial vendor (Rockland) with recombinant SP_1505 via the following schedule. Rabbit was immunized via intradermal route with 0.1 mgs SP_1505 with Complete Freund's Adjuvant (CFA) followed by an intradermal 0.1 mg booster injection with Incomplete Freund's Adjuvant IFA as an adjuvant at day 7, followed by two subcutaneous 0.1 mg booster injections at days 14 and 28 with IFA. Terminal bleed was collected on day 52 following challenge. SP_1505 IgG was purified from immunized rabbit serum using protein G resin and columns (Pierce) according to manufacturer specifications. Following purification, antibody was concentrated using 10,000 MWCO centrifugal filters (Millipore) and was dialyzed three times against PBS in a 3.5 kDa Slide-A-Lyzer dialysis cassette (Thermo Scientific). Antibody specificity was determined by western Blot against the parental wild-type and isogenic SP_1505 mutant, performed in duplicate as indicated in Fig. 4. Cellular fractionation of the wild-type and isogenic mutant was used to confirm the cross-reactive band localized to the membrane fraction, as predicted for the SP_1505 protein. No cross-reactivity was observed for the mutant in any of the blots.

### Cell fractionation, TCA precipitation and western blotting.
Strains were grown in Todd-Hewitt broth to OD 0.4. Following this, cells were fractionated as previously described in ref. [81]. Briefly, 2 mL of culture was centrifuged at maximum speed. The pellet was resuspended in cell wall digestion buffer [1× Protease inhibitor cocktail (Roche), 300 U/µL mutanolysin, 1 mg/mL lysozyme in a 30% sucrose-10mM Tris (pH 7.5) buffer with 20 mM MgCl2 and 20 mM MES (pH 6.5)] and incubated at 37 °C for 60 min. After centrifugation, the supernatant containing the cell wall was saved. Pelleted protoplasts were snap frozen in a dry ice ethanol bath, then treated with MgCl2, CaCl2, DNase I (Qiagen), and RNase A (Roche) in 50 mM Tris buffer (pH 7.5) with 20 mM HEPES (pH 8.0), 20 mM NaCl, and 1 mM DTT with protease inhibitors. The pellet was incubated on ice for 1 h, then spun at max speed for 30 min at 4 °C. The supernatant, which contained the cytoplasmic fraction, and the pellet, which contained the membrane fraction, were saved. 100%

TCA was added to the samples so that the final concentration of TCA was 20%. Samples were incubated on ice for 30 min, then centrifuged at full speed at 4 °C to pellet precipitated protein. The TCA supernatant was aspirated, and the pellet was washed twice with 100% acetone, then air-dried at 95 °C for 1 min. Pellets were resuspended in NuPage LDS sample buffer (Thermo Scientific) and boiled at 100 °C for 10 min. Samples were loaded into NuPage SDS-PAGE gels (Thermo Scientific) and transferred to nitrocellulose membranes using the XCell Sure-Lock mini-cell electrophoresis system (Thermo Scientific). Nitrocellulose membranes were blocked overnight in 5% NFDM and treated with primary antibody against SP_1505 at a concentration of 1:500. After washing, membranes were treated with secondary antibody goat anti-rabbit IgG-HRP (Bio-Rad catalog #1721019) at a concentration of 1:3000. Membranes were developed using the SuperSignal West Dura Extended Duration Substrate (Thermo Scientific) and were visualized using a BioRad ChemiDoc MP imaging system.

### Antibiotic-antibody targeted in vitro bacterial survival.
Bacteria were inoculated from TSA plates into C + Y media, at OD 0.4, culture was split into 1 mL aliquots and treated with vancomycin (0.25 µg/mL) or daptomycin (0.5 µg/mL). For antibody treatment, strains were grown in C + Y media until OD 0.3. At this time, samples were treated with SP_1505 antibody or control rabbit IgG antibody (Sigma) at concentrations indicated in figure legends, incubated for 30 min, followed by antibiotic treatment. At 4 h post-antibiotic addition samples were plated for bacterial enumeration.

### Antibiotic-antibody mouse challenge.
Isoflurane-anesthetized 7-week-old female BALB/c mice were inoculated intranasally with $10^6$ CFU of wild-type pneumococcal cells in a volume of 100 µL. Eight hours following the challenge mice were treated with vehicle (Plasmalyte), vancomycin (0.25 mg/kg), daptomycin (2.5 mg/kg), a-SP_1505 antibody (100 uL), and control rabbit IgG. At 16 h following antibody/antibiotic treatment (24 h post challenge) mice were euthanized, and lungs and chest cavity blood were removed for quantification of bacteria. Whole lungs were washed twice in PBS, and lung tissue was subsequently homogenized in 1 mL PBS. Homogenized lung samples were centrifuged at 300xg, and bacteria-containing supernatant was plated onto Neomycin-containing blood agar plates for CFU titers. Mouse experiments were approved under St. Jude Children's Research Hospital IACUC approved protocol #538 and Boston College IACUC approved protocol #2019-007-01. Mice were housed with a 12 h/12 h:dark/light cycle. The room temperature set point was 71 degrees F (±2 degrees) and the humidity setpoint was 40%.

### Peptide production.
Peptide P1 (Ser-Asn-Gly-Leu-Asp-Val-Gly-Lys-Ala-Asp) and peptide P2 (Ala-Lys-Thr-Ile-Lys-Ile-Thr-Gln-Thr-Arg) were synthesized on a preloaded Wang resin using the standard Fmoc/tBu chemistry for peptide synthesis. All coupling reactions were carried out in DMF using HBTU as the coupling reagent, 0.4 N-Methyl Morpholine in DMF as base. After each coupling, deprotection of the Fmoc group was done by using 20% piperidine in DMF. After completion of synthesis, peptides were cleaved from resin using TFA and purified using RP-HPLC. The integrity and purity of the peptides were confirmed using LC–MS.

### Antibiotic accumulation.
Antibiotic accumulation was determined as previously described[82]. *S. pneumoniae* were grown in THY to OD 0.6. Cells were pelleted, washed twice in PBS and resuspended in 3.5 mL PBS. 1 mL of cells were incubated with 50 µM antibiotic for 10 min at 37 °C. Following incubation, 800 µL of drugged cells were spun (3 min, 13,000xg) through 700 µL of a 9:1 mix of AR20 and high temperature silicon oils (cooled to −80 °C), after which the supernatant of silicone oil and free compound were carefully removed. For lysis, pelleted cells were resuspended in 200 µL dH2O and lysed via bead beating (3x 15 s at 5 m/s). Debris was pelleted (10' at 20,000xg) and 100 µL of supernatant was removed and saved. Cell debris was resuspended in the remaining 50 µL dH2O and mixed with 200 µL methanol. Potential cell debris was pellet again and 150 µL of the methanol extract was mixed with the 200 µL dH2O supernatant from the previous step. The extract was pelleted one final time (10' at 20,000 g) before being filtered (0.22 µm).

Samples were analyzed with a Waters Acquity M Class series UPLC system and Xevo G2 QTOF tandem MS/MS with Zspray. Hundred nanoliters of extract was separated using a Phenomenex Kinetex 2.6 µm XB-C18, 100 Å (300 µm × 150 mm) column with solvent A, 0.1% formic acid in water, and solvent B, 0.1% formic acid in acetonitrile. The inlet method utilized a flow rate of 8 µL min$^{-1}$ with the following gradient: 0−4 min, 99.9% solvent A and 0.1% solvent B; 4–5 min, 10% solvent A and 90% solvent B; 5–6 min, 99.9% solvent A and 0.1% solvent B. Tandem mass spectra were acquired with the following conditions: Ciprofloxacin: CV:20, CE:25, m/z ion: 333.14.→245.11; Kanamycin: CV:40, CE:20, m/z ion: 485.25→163.11. High-resolution spectra were calibrated by co-infusion of 2 ng mL$^{-1}$ leucine enkephalin lockspray (Waters). Data were quantified using Waters MassLynx software v4.2 where the AUC was determined by integrating the corresponding daughter peak of the parent compound. Concentrations of the unknown compounds were determined by the linear fit of the corresponding standard curve. Concentrations are reported as the average of three biological replicates.

**(p)ppGpp induction and LC/MS analysis**. *S. pneumoniae* strains were grown at 37 °C in 10 mL ThyB to an OD of ~0.5. Cultures were split into 5 mL aliquots for mupirocin-treated versus untreated controls. To induce the stringent response and ppGpp production, mupirocin was added in a final concentration of 25 µg/mL and incubated at 37 °C for 30 min. Cells were centrifuged at $6000 \times g$ for 5′, supernatant was discarded and cell pellets were frozen at −80 °C. For LC/MS analysis cell pellets were resuspended in 2 mL cold methanol, and 150 pmol of $[^{13}C_{10}]$-GTP (Sigma) was added and incubated at −80 °C for 30 min. Samples were centrifuged at $4000 \times g$ for 10′, and the supernatant was removed and dried overnight in a Savant Speedvac Concentrator SPD 1010 (Thermo Fisher). Samples were analyzed using a Shimadzu Prominence UFLC attached to a QTrap 4500 equipped with a Turbo V ion source (Sciex). Samples (5 µL) were injected onto a SeQuant ZIC-cHILIC, 3 µm, $2.1 \times 150$ mm column at 30 °C (Millipore) using a flow rate of 0.3 mL/min. Solvent A was 25 mM ammonium acetate, and Solvent B was 75% acetonitrile + 25 mM ammonium acetate. The HPLC program was the following: starting solvent mixture of 0% A/100% B, 0 to 2 min isocratic with 100% B; 2 to 4 min linear gradient to 85% B; 4 to 17 min linear gradient to 65% B; 17 to 22 min isocratic with 65% B; 22 to 25 min linear gradient to 100% B; 25 to 30 min isocratic with 100% B. The QTrap 4500 was operated in the negative mode, and the ion source parameters were: ion spray voltage, −4500 V; curtain gas, 30 psi; temperature, 400 °C; collision gas, medium; ion source gas 1, 20 psi; ion source gas 2, 35 psi; declustering potential, −40 V; and collision energy, −40 V. The MRM transitions are: ppGpp, 602.0;159.0; pppGpp, 682.0;159.0, and $[^{13}C_{10}]$-GTP, 522.0; 159.0. $[^{13}C_{10}]$-GTP was used as the internal standard. The system was controlled by the Analyst softwarev1.7 (Sciex) and analyzed with MultiQuant™ 3.0.2 software (Sciex). Peaks corresponding to ppGpp and pppGpp were quantified relative to the internal standard. The limit of detection for ppGpp and pppGpp is 5 pmol, and for GTP, GDP, ATP and ADP 0.05pmol.

**In vivo mouse competition experiment wo/w antibiotics**. In vivo competition experiments were essentially performed as previously described (van Opijnen and Camilli 2012). Specifically, groups of at least 12 outbred 4-6-week-old Swiss Webster mice (Taconic Inc.,) were anesthetized by isoflurane inhalation and challenged intranasally (i.n.) with 50 µL, ~$1.5 \times 10^7$ CFU, bacterial suspension in 1× PBS. Each bacterial suspension contained a 1:1 mixture of *S. pneumoniae* TIGR4 wild type and ΔSP_0829 or ΔSP_1396. The challenge dose was always confirmed by serial dilution and plating on blood agar plates. Infected mice receiving antibiotic treatment were administered either 1 mg/kg cefepime (WTvsΔSP_0829) or 10 mg/kg meropenem (WTvsΔSP_1396) 16 h post-bacterial challenge by intraperitoneal (i.p.) injection. Antibiotic dosing was previously determined to reduce bacterial loads 10-100-fold in vivo. Mice were euthanized by $CO_2$ asphyxiation at 6 h post-antibiotic administration (or 22 h post-bacterial challenge). Blood by cardiac puncture, nasopharynx lavage, and total homogenized lungs were collected from each animal to determine bacterial burden by serial dilution and plating blood agar plates as previously described[18]. Mouse experiments were approved under St. Jude Children's Research Hospital IACUC approved protocol #538 and Boston College IACUC approved protocol #2019-007-01. Mice were housed with a 12 h/12 h:dark/light cycle. The room temperature set point was 71 degrees F (±22 degrees) and the humidity setpoint was 40%.

**Clinical-strain stop-codon analysis**. Four gene sets were compiled to test for the differential occurrence of stop codons in patient samples. Each gene set consists of 34 genes and are defined as: Set 1 consists of genes that when disrupted lead to a significant decrease in antibiotic sensitivity in the presence of at least one antibiotic (in vitro ABx fitness positive), and have no fitness defect in lung and nasopharynx (in vivo neutral or positive); Set 2 consists of genes that when disrupted lead to a significant decrease in antibiotic sensitivity in the presence of at least one antibiotic (in vitro ABx fitness positive), and have a significant fitness defect in lung and nasopharynx (in vivo fitness negative); Set 3 consists of genes that when disrupted have no fitness benefit in any of the antibiotics (in vitro ABx fitness neutral), but with a significant fitness benefit in lung and nasopharynx (in vivo fitness positive); Set 4 consists of genes that have decreased fitness in the presence of antibiotics (in vitro ABx fitness negative), and that have a significant fitness defect of >15% in lung and nasopharynx (in vivo fitness negative). The PATRIC database (https://www.patricbrc.org/) was screened for antibiotic-resistant *S. pneumoniae* isolates. There is a potential risk that isolates in the database are clonally related, which could mean that multiple isolates would contain exactly the same sequence and for instance the same stop codon, which could bias the analysis. To reduce this potential bias candidate isolates were limited to those belonging to a different MLST type. While this considerably reduced the number of potential isolates, we were able to collect 533 β-lactam resistant and 1147 co-trimoxazole resistant strains. Moreover, an equal number of non-resistant strains were compiled. From each genome, gene sequences were extracted that match those from each of the 4 gene sets. Each gene was scanned for premature stop codons occurring in the first 90% of a gene. For each gene set the number of strains with at least one stop codon in the gene set were recorded, as well as the total number of stop codons in all genes in a set. To test for differences in the number of isolates containing a stop codon within (susceptible vs. resistant) and between sets a Fisher's exact test was performed.

**Reporting summary**. Further information on research design is available in the Nature Research Reporting Summary linked to this article.

## Data availability

Sequencing data generated in this study is available at the Short Read Archive (BioProject accession number PRJNA750080). Tn-Seq fitness data used across the manuscript can be found in Supplementary Data 2. Growth and tolerance data can be found in Supplementary Data 8 and Purine Metabolism data in Supplementary Data 9. Gene annotation information was obtained from PATRIC (https://www.patricbrc.org/), GenomeNet (https://www.genome.jp) and UniProt (https://www.uniprot.org/). Gene sets were assembled from sequences obtained from PATRIC. Source data are provided with this paper.

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

## Acknowledgements

DNA sequencing was performed at the Boston College Sequencing Core. The authors wish to thank Jon Anthony for running the *Aerobio* sequencing analyses platform, and

Ralph Isberg and Vaughn Cooper for valuable discussions. This work was supported by a Charles King Trust Fellowship to F.R., NIH R01s GM124231 to J.G., AI110724 and AI148470 to T.v.O., and U01 AI124302 to T.v.O. and J.W.R.

## Author contributions

T.v.O. devised the study and wrote the manuscript. E.R., B.S., L.M.N.R., F.R, A.T.N., S.J.W., B.J., N.B, K.L., J.G., M.F., S.M.R., R.E.L., C.O.R., J.W.R., and T.v.O. performed wet-lab experiments, data collection and interpretation. D.L. and T.v.O. performed Tn-Seq data analysis, pathway and network construction, analysis and interpretation. J.W.R. contributed to key conceptual ideas. All authors contributed to manuscript editing and approved the final paper.

## Competing interests

The authors declare no competing interests.
