## [Peer Review File · Nature Communications]

A genome-wide atlas of antibiotic susceptibility targets and pathways to toleranceREVIEWER COMMENTS

Reviewer #1 (Remarks to the Author):

The work by Leshchiner et al. provides an extensive profile of drug susceptibility determinants under antibiotic treatment. The authors carry out a Tn-seq experiment with *Streptococcus pneumoniae* under treatment with 20 diverse antibiotics and build a genome-wide interaction map. They then demonstrate that this genome-wide atlas can be harnessed for several applications. For instance, they identify effective adjuvant targets, from which they develop a combinatorial antibody-antibiotic treatment that is effective in a mouse model. The interaction map also reveals new information about downstream effects that lead to antibiotic lethality, mainly that defects in the production of the alarmone (p)ppGpp causes lowered ATP synthesis and reduced antibiotic susceptibility. Finally, the authors demonstrate clinical relevancy by showing that genetic targets that decrease sensitivity are found in clinical samples.

While this could be an important resource of interest to the broad readership of Nature Communications, there are several statements made in the text that are not fully supported by the data. In particular, the authors claim that this is an atlas of pathways to tolerance, even though their Tn-seq experiment was not designed to probe for tolerance. Their use of the terms “tolerance” and “susceptibility” is also confusing as they do not provide MIC measurements in accordance with the field standards on distinguishing between tolerance and resistance (Brauner 2016 Nature Reviews Microbiology). Additionally, more discussion is needed to place this work in the context of the existing literature.

Major comments:

1. The authors claim that their “pathway analysis leads to a genome-wide tolerome” (line 38). However, their Tn-seq experiment was conducted under antibiotic concentrations that slow growth by 30-50%. If these concentrations are sub-MIC, then they would not enrich for insertion mutants that are tolerant, which is a killing- and not growth-based phenotype. Though the authors demonstrate that some of their targets that have reduced antibiotic susceptibility under the Tn-seq experimental conditions do survive antibiotic lethality through CFU counting and MDK determination, that is not what the Tn-seq results are directly probing. The authors should adjust their claims that this analysis reveals the tolerome.
2. Throughout the paper the authors loosely apply the words “susceptibility” and “tolerant” without the proper evidence. In particular, the paper does not provide MIC data in addition to killing enumeration, which is the standard for distinguishing between different types of antibiotic survival. Without the requisite MIC data, it is unclear whether strains that have increased survival are in fact tolerant, or whether they are resistant. The authors should provide MIC data for any strains from the Tn-seq experiment that they examine in further detail.
3. The antibiotic Tn-seq experiments are supplemented with additional Tn-seq data taken in 17 additional environments, which together are used to create the genome-wide interaction map. However, the majority of the 17 additional environments listed in the supplement are non-stressful conditions, mostly the addition of sugars. The authors should clarify why these additional conditions were included to construct the genome-wide map under antibiotic treatment.
4. Important information is missing from the description of the main Tn-seq experiment in the Methods, including the time of antibiotic exposure, cell density, and how they define slow growth

(growth rate?). The authors should expand on these methods.

5. There are several areas where the authors should refer to published work in order to give context to this advancement. In particular:

a. The tolerome: what other works have explored genes that contribute to tolerance on a large scale? The authors should refer to Brauner et al Nature Reviews Microbiology 2016 and Erickson et al ACS Synthetic Biology 2017.

b. Genome-wide mapping of antibiotic susceptibility determinants: how does this work expand upon other methods of genome-wide mapping?

c. Downstream effects that contribute to antibiotic lethality: the authors should refer to Yang et al Cell 2019, which also explores the effects of purine metabolism on antibiotic lethality.

Minor comments:

1. Lines 89-93: the authors state that antibiotic tolerance and persistence are induced upon antibiotic exposure. There are many cases where tolerance and persistence are not antibiotic-induced but rather an effect of the environment, such as biofilms and low-nutrient conditions. The authors should clarify what they mean by antibiotic induction of tolerance.

2. The readability of lines 177-233 could be improved. As this section is mostly a list of pathways that contribute to antibiotic susceptibility, this could be summarized in a table.

3. The data in Figure 1C is too small, please enlarge this.

Reviewer #2 (Remarks to the Author):

Summary

In their paper, Leschiner et al. build a thorough chemical-genetic network in *Streptococcus pneumoniae*. The authors apply Tn-Seq in *S. pneumoniae* to 20 antibiotics, 17 non-antibiotic environmental conditions (data gathered in a prior study), and two in vivo infection settings. The first section of the paper develops a genome-wide view of antibiotic susceptibility, defining some key gene-drug interactions that are explored in more detail later in the paper. Section two explores the relationships of genes in the interaction network by analyzing which genetic fitness profiles are similar across conditions, and which are negatively correlated. This resulted in clustering of functionally related genes of known and unknown function. Section three discussed a wide array of antibiotics and genes, highlighting various positive and negative interaction profiles with a focus on clusters of genes in seven biological pathways. This broad exploration led to the fourth section of the paper, which focused on the interaction between *cozEb* and (daptomycin or vancomycin). This in vitro interaction was leveraged to make an antibiotic (daptomycin) + anti-*cozEb* antibody combination for an in vivo nasal/lung infection. Section five focused on genetic perturbations resulting in antibiotic tolerance, specifically the promiscuous tolerance afforded by inhibition of *amiE* and *amiC*. Indeed, the authors show that deletion of *amiE* results in gentamicin tolerance. Section six highlights the altered antibiotic susceptibility profiles (sensitivity and tolerance) that result from genetic manipulation of purine metabolism, the stringent response, and ATP biosynthesis. Much prior work has explored the importance of

these pathways in bactericidal antibiotic efficacy. The final section first validated two genes (deoB and pstA) as conferring tolerance in vivo, then attempted to predict tolerance-conferring mutations from genomes of clinical isolates.

Comments

This is a complex study that should be published after the revisions outlined below. I appreciate how much work went into building out this manuscript and the authors should be commended. The paper was organized logically, showcasing the dataset (Figures 1 and 2), then using this dataset to explore varying drug-gene interactions (Figure 3 and Figure 6) in the context of both antibiotic sensitivity (Figure 4 and Figure 6) and antibiotic tolerance (Figure 5), then finally investigating tolerance in vivo (Figure 7 and Figure 8). However, it is a long paper, and I found myself at times losing focus; it would serve the paper well, in my opinion, to remove Figure 2, remove Figure 8, and combine Figure 3 and Figure 6.

Figure 2 – this figure (and accompanying text) goes in-depth to explain the identification of functional relationships amongst genes in their dataset. This is necessary for the discussions that follow, but to me this feels like a supplementary discussion. I found it a little distracting.

Figure 8 – this is the weakest part of the study in my opinion. From my understanding, the authors attempt to identify antibiotic-tolerant clinical isolated by finding stop codons within tolerance-conferring genes from their Tn-seq dataset. This is fine, but worryingly reductionistic. Indeed, there are a wide array of mutations that can inhibit the function of some important gene that leads to tolerance. Given the limitations inherent to this type of in vitro to in vivo search, especially in a genomic space as wide as that which may confer tolerance, I would remove this figure altogether. To me, it adds little to the study and may in the worst case be misinterpreted by readers without expertise in the field.

Figure 3 and 6 – These feel like related figures. Figure 3 touches on chemical-gene interactions across an array of biological processes. Figure 6 goes into more detail of antibiotic - [purine/(p)ppGpp/ATP] interactions, which has been explored previously. Since both figures highlight the utility of their Tn-seq database to reveal both suppressive and enhancing drug-gene interactions, merging them serves the purpose to simplify the narrative and decrease the length of the paper.

One concern that I have is that most of the follow-up experiments investigating tolerance do not show full kinetic kill curves – the only way to define tolerance. For every instance in the paper that discusses gene disruption conferring antibiotic tolerance, it is not sufficient to show growth curves. Rather, the authors should be performing kinetic kill curves and quantifying the differential survival of wild type cells and mutant cells. I appreciate that this is done in Figure 5D and (somewhat) Figure 7B, but there are other instances of tolerance that are not shown in the best manner (Figure 6 C/D, for example). Similarly, when discussing enhancement of antibiotic activity upon gene disruption, it would be ideal to show decreases in antibiotic MIC (if present). Even if not, this would help readers understand the amplitude of the antibiotic-enhancement phenotypes of each drug-gene interaction discussed in detail in the paper.

Figure 7A and 7C are unclear to me even after a couple reads. They require better explanation.

Is there an online and searchable database so that this dense resource can be made available to the scientific community? This would be a valuable contribution.

Reviewer #3 (Remarks to the Author):

This manuscript describes results from a very extensive genome-wide interrogation of antibiotic sensitivity and resistance/tolerance in the pneumococcus. Building on their expertise, the authors subjected 6 independent Tn-seq libraries to treatment with 20 antibiotics belonging to four different classes. They combined these new results with similar results in other conditions, including in vivo Tn-seq screens in mice, to generate networks of gene interactions and identify genes/operons/pathways/regulatory networks that mediate sensitivity and/or resistance/tolerance, and make predictions as to how genes of unknown function might be linked to or influence genes/pathways of known function, and how in vitro data would translate in vivo.

The body of data generated, and the extensive analyses presented are significant. Selected cases of increased and decreased sensitivity are confirmed with a suite of very elegant and extensive experimental procedures. These highlight new mechanisms associated with drug responses in the pneumococcus.

The overarching goal of the study is to identify novel ways of rescuing drug sensitivity but also identify signals/features that could suggest the emergence of resistance. Overall, while selected examples make a strong case towards the goal, the study does not provide an approach or a tool to predict emergence of resistance. The opportunity to use the data for in silico modeling is not implemented or discussed.

A one-stop list of candidate “precursors” of resistance, and of candidate mediators of persistence, is desired. It is hard to gather from the large amount of supplemental data. The authors should provide a detailed guide to the content of the latter. Along such lines, but conversely, the overview of the study in panel A of Figure 1 is informative.

The many Figures presented are beautiful but also really dense and complex. Many undefined abbreviations, acronyms, and truncated words are used throughout, making it hard to intuitively understand the message conveyed. A striking example, that pertains to the figures and the main text, is that only one of the four classes of antibiotics, “CWSI”, is defined. Such terms should be introduced/defined for the general readership.

Line 36: change “generate” to “generated”.

Line 170: replace “))” by “)”.

Figure 3 a and b: the x-axes should be better defined. How do “effects” compare to “phenotypes” (used in the text) vs. “instances”?

Dear Editor and Referees,

We thank you for reviewing our manuscript and investing your valuable time to help us improve this work. We were really excited to see that you were overall very positive and appreciated the work we have put into developing this research, the overall clarity of the manuscript and the aesthetics of the figures. We want to ensure you that we have taken all of your comments very seriously and have made changes throughout the manuscript to clarify ideas, concepts and results, we have reduced complexity in certain sections, and we have added more data/results where this was necessary/requested. We have addressed each comment below in as much detail as we could and we have motivated the changes that we made and where we made them in the manuscript. We have also added a document with track changes so it will be easier to see what we changed exactly. We hope you like and appreciate the changes we made and that we will be able to share this work with the community as soon as possible.

Sincerely, and on behalf of all the authors,
Tim van Opijnen

Reviewer 1. - Major comments:

Reviewer 1 - Comment 1. The authors claim that their “pathway analysis leads to a genome-wide tolerome” (line 38). However, their Tn-seq experiment was conducted under antibiotic concentrations that slow growth by 30-50%. If these concentrations are sub-MIC, then they would not enrich for insertion mutants that are tolerant, which is a killing- and not growth-based phenotype. Though the authors demonstrate that some of their targets that have reduced antibiotic susceptibility under the Tn-seq experimental conditions do survive antibiotic lethality through CFU counting and MDK determination, that is not what the Tn-seq results are directly probing. The authors should adjust their claims that this analysis reveals the tolerome.

Author response:

The execution of Tn-Seq experiments, data analysis and interpretation commented on by the reviewer is important and before we highlight in detail how we have addressed the reviewer's comments in the manuscript, we first describe some important considerations and background information concerning the manner in which we perform Tn-Seq experiments and data analysis, which we believe helps put our changes to the manuscript in the proper context.

Explanation concerning overall Tn-Seq approach and how to interpret fitness data (e.g., altered ABX sensitivity, positive fitness, increased relative growth and tolerance):

Over the last 10+ yrs we have established a clear Tn-Seq protocol that aims to extract as much, and as detailed information from a Tn-Seq experiment as possible. This means we follow how a population (Tn-Seq insertion library) develops and expands over the course of the experiment, irrespective whether the experiment is performed *in vitro* or *in vivo*. Moreover, a selective environment/condition is always developed/applied in such a way that it can be directly compared to a control condition; i.e., it preferably only has a single variable, for instance a single different carbon source, or lack of an amino acid. Under such circumstances you can ask the question which genes/genetic regions are involved in dealing with the altered variable. For instance, you may see a transporter or pathway that was previously inconsequential become critical under the new condition. In the case of a condition where a variable is added, such as an antibiotic, you want to create a situation where you affect growth significantly (e.g., 30-50% reduction), but not in such a way that you prevent all growth. This is important for at least 2 reasons: 1) A significant reduction within the right window creates a selective pressure that ensures you can identify genetic factors along a wide dynamic range that are important in the specific condition; 2) A condition that is too harsh, for instance one that abolishes growth, introduces an enormous amount of stochastic variation in an experiment, making it extremely hard to identify, with high certainty, whether a genetic factor is important in response to the selective pressure or not.

Importantly, we have previously shown that if you create a selective condition, such as an antibiotic (ABX) and you ramp up the pressure by increasing the ABX concentration than you increase the number of genetic factors that interact with the environment. This means that with a low selective pressure you can identify the factors that have the strongest effect on the phenotype, while if you ramp up the pressure you can start identifying genetic factors that have more subtle phenotypes (while also still catching the large effect ones). Moreover, by sampling over an extended period that a population develops/grows you identify genetic factors that are important across that entire range. This means that some genetic factors that you identify will affect early growth, others may affect late stage growth, while yet others may affect the entire life cycle. Thereby when it comes to an environment that reduces growth and is tunable, a 30-50% reduction is right in the sweet spot of identifying the largest number of genetic factors with the highest confidence.

Additionally, we have invested a large amount of time and effort in developing an analytical approach that translates the sequencing data from a Tn-Seq experiment into a quantitative fitness effect for each disrupted genetic region/gene in the library. Specifically, this means we fit the sequencing data and population expansion to an exponential growth model and can thereby calculate how a gene disruption affects a mutant's relative growth rate. For instance, if we find a gene disruption that results in a fitness of 0.5, it generally means the mutant grows twice as slow as the wild type. Importantly, because we take so many measurements, we can calculate a disruption's average effect on growth, its variance and standard deviation, enabling statistics to unequivocally predict whether a fitness effect is significant. These considerations with respect to experimental set-up and fitness assessment are worked out and described in detail in over a dozen publications including three detailed review articles in Nature Reviews Microbiology, Nature Reviews Genetics and Annual Reviews of Genetics¹⁻⁴.

With the experiments for the current manuscript it was initially a key goal to identify the genetic factors that are involved in overcoming stress triggered by antibiotics. We thus aimed to focus on those genes that significantly decrease fitness in the presence of an ABX, meaning a transposon disruption would increase ABX susceptibility. As described above, when you identify such a genetic factor it may be initially unclear what the exact phenotypic effect is, besides that it somehow reduces growth. By doing follow-up experiments and/or by placing it into a genome-wide framework, it is possible to elucidate when the phenotype occurs and thereby gain insight into why it occurs. Importantly, due to the sensitivity of our entire approach, besides negative effects on fitness (i.e., increased ABX susceptibility) we also identified many positive effects on fitness. We have seen such positive effects in most/all of our previous Tn-Seq projects, however, we (and neither have others) have never invested sustained effort into figuring out how to interpret these positive effects and determine what they actually mean/how important they are. In this current manuscript we did put in extensive effort to determine the importance and origin of these positive effects. First, like we show that a negative fitness effect quantitatively predicts increased ABX sensitivity resulting in decreased relative growth, we reasoned that a positive effect would indicate decreased ABX sensitivity. Additionally, we reasoned that this decreased sensitivity could include at least two phenotypes: 1) Increased relative growth in the presence of the ABX compared to the WT; 2) Increased relative survival in the presence of the ABX compared to the WT.

Therefore, we measured both; i.e., growth and survival. We show in Figure 7A that many of the mutants have an increased relative growth rate in the presence of an ABX compared to the WT. Additionally, we show in Figure 7B that many also have increased survival in the presence of an ABX compared to the WT, while most mutants display both. This means that mutants that trigger positive fitness effects indeed display decreased ABX sensitivity, which is exhibited as: 1) Increased relative growth; 2) Increased survival; or 3) Both. Positive fitness effects thus include a range of decreased ABX sensitivity phenotypes, of which a substantial subset are tolerance mutants (i.e., those with increased survival).

Changes made to the manuscript in relation to Reviewer 1 comment 1:

In the manuscript we have made changes to highlight some of the considerations described above to better illustrate how to interpret the fitness data, how the data leads to specific (validate) phenotypes, and to specifically address the reviewer's concerns:

1. We now highlight that while Tn-Seq experiments are performed under ABX selection that maintains growth, our detailed analysis and validation experiments illustrate how increased fitness can be interpreted. We thereby now clearly define on **P6L145**, **P7L212-219**, **P9L357-358**, **P10L361-368**, **P11L408-411**, **P13L485-502** what decreased and increased fitness means and how validation experiments show that this translates into different phenotypes; i.e., decreased fitness means increased

ABX sensitivity, which is phenotypically expressed as decreased relative growth, while increased fitness means decreased ABX sensitivity, which is phenotypically expressed as increased relative growth in the presence of an ABX, and/or tolerance. Furthermore, we explain in detail how fitness is calculated on **P17L701-747**, and how this can be translated into certain phenotypes.

2. To better illustrate the phenotypes we tried to better explain the patterns of altered ABX sensitivity in pathway/process Figure 3 and Supplementary Figures 2,3, which highlight in detail on a genome wide level how inhibition of a pathway/process can lead to lowered ABX sensitivity. Importantly, on **P13L485-502** we improved explanations of Figure 7A and B and how they serve as a relatively large-scale validation of increased fitness phenotypes; i.e., it shows that inhibition of specific pathways indeed leads to decreased ABX sensitivity, and indeed increased relative growth, and/or increased survival (tolerance). Additionally, we have also added the MICs in Supplementary Table 1 (but see comment on MICs and our response below).

3. Finally, we clarified throughout the manuscript including the conclusion section that our genome-wide analysis, the consistency in the pathway patterns on altered drug sensitivity, and our detailed validation data, show that within this large-scale identification of altered drug susceptibility, several key pathways exist that when inhibited lead to tolerance. The dataset thereby truly describes a genome-wide atlas of altered drug susceptibility, including pathways to tolerance. We get the sense from the reviewer that they think that readers may interpret that the entire decreased ABX sensitivity dataset thereby describes the tolerome. We agree that this could be a bit confusing and is not what we want. Therefore, throughout the manuscript including the abstract we now clarify that the dataset reveals a genome-wide atlas of cellular processes that can lead to altered drug susceptibility, and that the part of the Atlas that depicts decreased ABX sensitivity (i.e., increased fitness) includes a genome-wide 'tolerome', composed of a wide variety of pathways and processes that when modulated trigger tolerance *in vitro* in an ABX dependent manner.

Reviewer 1 - Comment 2. Throughout the paper the authors loosely apply the words "susceptibility" and "tolerant" without the proper evidence. In particular, the paper does not provide MIC data in addition to killing enumeration, which is the standard for distinguishing between different types of antibiotic survival. Without the requisite MIC data, it is unclear whether strains that have increased survival are in fact tolerant, or whether they are resistant. The authors should provide MIC data for any strains from the Tn-seq experiment that they examine in further detail.

Author response:

The reviewer is right, and as indicated above we have added the MICs in Supplemental Table 1 and how they were determined in the methods section. We also now refer in multiple places in the manuscript where we talk about tolerance to the MICs of the mutants. This shows that the mutants that display increased survival, mostly show no to little change in MIC, confirming their increased tolerance phenotypes. Additionally, we now clearly define in multiple sections throughout the manuscript including the results, conclusion and methods sections what we mean with ABX susceptibility. We thereby explain and validate that decreased fitness means increased ABX sensitivity, which is phenotypically expressed as decreased relative growth, while increased fitness means decreased ABX sensitivity, which we show can translate into: 1. Increased relative growth in the presence of an ABX; 2. Tolerance, or 3. Both.

Reviewer 1 - Comment 3. The antibiotic Tn-seq experiments are supplemented with additional Tn-seq data taken in 17 additional environments, which together are used to create the genome-wide interaction map. However, the majority of the 17 additional environments listed in the supplement are non-stressful conditions, mostly the addition of sugars. The authors should clarify why these additional conditions were included to construct the genome-wide map under antibiotic treatment.

Author response:

On **P6L160-169** we now better explain why the additional conditions are added to the analysis, which is to increase the resolution of the interactions that can be discovered. Basically, it increases the statistical power to discover whether genes and/or pathways interact and/or are associated with each other. Importantly, it may be a bit of a misunderstanding or possibly semantics what you call a stressful condition. As described above in depth, Tn-Seq and many other screening tools, can be used to determine which genes are involved in dealing with a specific condition/environment. This can be antibiotics, or this can be a different carbon source (identifying

the pathways involved in processing that sugar, and those pathways that may make critical use of that sugar). Alternatively, it can be a missing amino acid, or a metal ion. You thereby can't realistically say that one is a stress and another is not. For instance, as we have shown before, processing sialic acid as the sole carbon source puts a lot of stress on a wide variety of systems that then all of a sudden are critical, while if glucose is the carbon source these same systems/pathways may be dispensable^{1,5-9}. Adding the additional conditions to the analysis thus helps in identifying those pathways and genes that interact.

Reviewer 1 - Comment 4. Important information is missing from the description of the main Tn-seq experiment in the Methods, including the time of antibiotic exposure, cell density, and how they define slow growth (growth rate?). The authors should expand on these methods.

Author response:

As we have described these and many more details and considerations in a considerable number of publications, and as mentioned above they are also detailed in three reviews, this may have resulted in us being a little bit scant on some important experimental and analytical details in this manuscript. We have added the requested information to the Methods section on **Pages 17 and 18** for instance indicating the timing of the experiments, the starting density, culturing generations, fitness and growth rate calculations.

Reviewer 1 - Comment 5. There are several areas where the authors should refer to published work in order to give context to this advancement. In particular:

a. The tolerome: what other works have explored genes that contribute to tolerance on a large scale? The authors should refer to Brauner et al Nature Reviews Microbiology 2016 and Erickson et al ACS Synthetic Biology 2017.

b. Genome-wide mapping of antibiotic susceptibility determinants: how does this work expand upon other methods of genome-wide mapping?

c. Downstream effects that contribute to antibiotic lethality: the authors should refer to Yang et al Cell 2019, which also explores the effects of purine metabolism on antibiotic lethality.

Author response:

We have expanded the conclusion section on **P15-16**, we have put our main results into a more detailed context, we have added relevant articles including those that the reviewer suggested, we better compare some of our main results to what others have found in other organisms, and we added explanations that should clarify how we and others will be able to use our data in the future, including for modeling purposes.

Reviewer 1 Minor comments:

1. Lines 89-93: the authors state that antibiotic tolerance and persistence are induced upon antibiotic exposure. There are many cases where tolerance and persistence are not antibiotic-induced but rather an effect of the environment, such as biofilms and low-nutrient conditions. The authors should clarify what they mean by antibiotic induction of tolerance.

Author response:

We tried to clarify the sentence on **P4L93-95** and added several references.

2. The readability of lines 177-233 could be improved. As this section is mostly a list of pathways that contribute to antibiotic susceptibility, this could be summarized in a table.

Author response:

We are trying to help the reader understand how Figure 3 and Supplementary Figures 2,3 should be read/interpreted. These figures contain a lot of information and it would be a shame if the meaning gets lost in translation. However, we do agree with the reviewer that the section is long and a bit of a laundry list so we significantly shortened and tried to make it easier to follow while retaining the key information (**P8L246-267**).

3. The data in Figure 1C is too small, please enlarge this.

Author response:

Data in Fig.1C has been enlarged.

Reviewer 2. Major comments:

Reviewer 2 - Comment 1.

This is a complex study that should be published after the revisions outlined below. I appreciate how much work went into building out this manuscript and the authors should be commended. The paper was organized logically, showcasing the dataset (Figures 1 and 2), then using this dataset to explore varying drug-gene interactions (Figure 3 and Figure 6) in the context of both antibiotic sensitivity (Figure 4 and Figure 6) and antibiotic tolerance (Figure 5), then finally investigating tolerance in vivo (Figure 7 and Figure 8). However, it is a long paper, and I found myself at times losing focus; it would serve the paper well, in my opinion, to remove Figure 2, remove Figure 8, and combine Figure 3 and Figure 6.

Figure 2 – this figure (and accompanying text) goes in-depth to explain the identification of functional relationships amongst genes in their dataset. This is necessary for the discussions that follow, but to me this feels like a supplementary discussion. I found it a little distracting.

Author response:

We understand what the reviewer is thinking here, but we believe Figure 2 is really important to show that the data + analyses are robust and lead to a genome-wide view of the genomic architecture, which is illustrated by intra and inter pathway/process clustering. Additionally, it shows it is possible to extract more intricate interactions as well those that have real biological value. Moreover, we explore several of these interactions throughout the paper. Another reason why we think the figure should remain is that we use illustrative details of the Figure 2 network in Figures 4, 5 and 6, which would make less sense without having seen the global network and some of its highlighted details in Figure 2. We thus really would like to retain this figure in the manuscript. However, if the editor agrees with the reviewer that the figure does not add anything substantial we can discuss how to remove it without losing the context of subsequent figures.

Reviewer 2 - Comment 2.

Figure 8 – this is the weakest part of the study in my opinion. From my understanding, the authors attempt to identify antibiotic-tolerant clinical isolated by finding stop codons within tolerance-conferring genes from their Tn-seq dataset. This is fine, but worryingly reductionistic. Indeed, there are a wide array of mutations that can inhibit the function of some important gene that leads to tolerance. Given the limitations inherent to this type of in vitro to in vivo search, especially in a genomic space as wide as that which may confer tolerance, I would remove this figure altogether. To me, it adds little to the study and may in the worst case be misinterpreted by readers without expertise in the field.

Author response:

There may be a bit of confusion here. In the section that includes Figure 8 we search for the presence of stop codons in clinical strains in genes we identified with Tn-Seq that may lead to lowered ABX sensitivity, and in many cases tolerance. We agree with the reviewer that there are many changes that could lead to alteration/inactivation of the function of a gene. However, we do not know which mutations could have such an effect, except for stop codons. What we mean with that is that a transposon insertion disrupts the function of a gene, and a stop codon is the only change we can be certain of to have the same/similar effect. Which is why we only look for premature stop codons in the clinical strains, and ignore other changes. The point we are trying to make with Figure 8, which we do think is worthwhile, is to show that there are a very large number of instances of premature stop codons in clinical samples that would cause lowered antibiotic sensitivity (as predicted by the Tn-Seq data). Moreover, many of the clinical strains that carry these premature stops are resistant to one or more ABXs. The importance of this is that it shows it would be worthwhile to look at this in more detail, i.e., mutations occur in clinical samples, and our data suggest that this could have direct consequences on their ABX susceptibility. Thereby mapping out mutations that occur outside of direct targets and determine their effects on

drug susceptibility, the occurrence of antibiotic treatment failure, and/or the occurrence of resistance, could result in a new way to predict/prevent treatment failure or resistance, and eventually even preemptively change treatment strategies. While we agree that this will not be easily done, our analyses are an argument that this could be well worthwhile. Moreover, our Tn-Seq strategy, data and analyses show that such an approach could be used as a way to more rapidly built such an understanding, i.e., it can be a starting-off point. We're not saying that this is the only way, but it could definitely hold value. Lastly, removing the entire figure and analyses would take away the argument to do this. However, we rewrote a part of that section to try to clarify some of these important points (P14L577-583).

Reviewer 2 - Comment 3.

Figure 3 and 6 – These feel like related figures. Figure 3 touches on chemical-gene interactions across an array of biological processes. Figure 6 goes into more detail of antibiotic - [purine/(p)ppGpp/ATP] interactions, which has been explored previously. Since both figures highlight the utility of their Tn-seq database to reveal both suppressive and enhancing drug-gene interactions, merging them serves the purpose to simplify the narrative and decrease the length of the paper.

Author response:

We understand where the reviewer is coming from. We do present a lot of data but with a resource such as this it is important to highlight the versatility of the approach and dataset. We could have opted to split the data up over multiple papers, but that would take away from the overarching message and the mineable aspect of generating a dataset like this. We carefully determined how to best present the data, construct an understandable narrative and logical flow to the paper, which the reviewer also highlighted in detail in their summary of the paper. We have thought about pushing Figure 6 up in the order of the paper and merging it with Figure 3, however this would affect the logical flow of the paper, change the narrative and result in a repeating of themes in different sections. As the reviewer has correctly noted our order is to show the reader in Figures 1-3 an overall overview of the experiments and data-set/resource we generated. Moreover, the goal is to show that there is merit to our data, i.e., there are patterns in the data that make sense with what we know, suggesting that if we find new things they may be true as well.

Figure 3 serves a second purpose in that it takes the entire data set and shows we can also zoom in on a pathway/process and identify patterns that are consistent throughout, further suggesting that we have a high confidence and valuable data-set/resource. Subsequently, we explore several themes in the manuscript in detail: **A)** In Fig.4. we explore whether we can use the resource and network to identify targets and design new proof-of-principle strategies; **B)** In Fig.5 we determine whether we can describe new biology. Moreover, we focus on what those positive fitness effects actually mean; showing they indicate lowered drug sensitivity, resulting in a better relative growth rate and/or tolerance. **C)** In Fig.6. we dive deeper into this question: we explore new biology, we explore the pathway/process with the most number of positive fitness results, and determine how these positive effects are linked to the output of the pathway. Specifically, we determine whether there is a link between decreased ABX susceptibility and the synthesis of (p)ppGpp, di/tri nucleotides including ATP. Importantly, we thereby also establish a link to neighboring and closely related pathways, which are all detailed in Supplementary Figs 2,3. **D)** In Fig.7 we then go on to show that decreased ABX sensitivity and tolerance are phenotypes that can be triggered by changes to many different pathways. Furthermore, by combining *in vitro* with *in vivo* data we find that a substantial number of changes would also cause lowered drug sensitivity/tolerance *in vivo*, and then finish with showing that our predictions can indeed be verified with *in vivo* experiments; i.e., mutants with lowered drug sensitivity *in vitro* display the same phenotype *in vivo*.

Importantly, if we would move Figure 6 up and merge it with Figure 3 it would change the narrative and disrupt the flow of the manuscript; i.e., we would talk about lowered drug sensitivity/tolerance repeatedly in non-consecutive sections and thereby in an order that would not gradually built but would jump around.

Reviewer 2 - Comment 4.

One concern that I have is that most of the follow-up experiments investigating tolerance do not show full kinetic kill curves – the only way to define tolerance. For every instance in the paper that discusses gene disruption conferring antibiotic tolerance, it is not sufficient to show growth curves. Rather, the authors should be performing kinetic kill curves and quantifying the differential survival of wild type cells

and mutant cells. I appreciate that this is done in Figure 5D and (somewhat) Figure 7B, but there are other instances of tolerance that are not shown in the best manner (Figure 6 C/D, for example). Similarly, when discussing enhancement of antibiotic activity upon gene disruption, it would be ideal to show decreases in antibiotic MIC (if present). Even if not, this would help readers understand the amplitude of the antibiotic-enhancement phenotypes of each drug-gene interaction discussed in detail in the paper.

Author response:

The kill curve data the reviewer is asking for are listed in Supplementary Table 8. We opted to highlight detailed data in Figure 5D, and summarize the tolerance data for all of the validated mutants by showing survival after 8 and 24hrs in Fig.7B. In Fig.7B we also included/summarized the survival data detailed in Fig.5D so the reader has a clear reference. Fig.7B also includes the tolerance data for the mutants shown in Fig6C/D. Importantly, like the reviewer mentions, we should have included the MIC data for these mutants, which we have now included in Supplementary Table 1 and we refer to the MICs in the manuscript when we talk about tolerance. Additionally, we have now more clearly defined what ABX sensitivity (decreased/increased) means phenotypically, we explain in more detail/clearly what the data in Figs 7A/B represent and how they are calculated, both in the main text and in the Methods section. As a result we believe this improves the overall ease to understand our figures and data and the reader should be able to get a comprehensive phenotypic overview of all validated mutants.

Reviewer 2 - Comment 5.

Figure 7A and 7C are unclear to me even after a couple reads. They require better explanation.

Author response:

We have clarified the description of these figures on **P13**, in the legend, as well as the description of data collection/analysis in the Methods section.

Reviewer 2 - Comment 6.

Is there an online and searchable database so that this dense resource can be made available to the scientific community? This would be a valuable contribution.

Author response:

We completely agree that this would be useful and we have previously done so for other datasets. However, as with many other 'industries' we have a personnel shortage and as of yet have not been able to complete such an online database. However, we are working on it and we hope to present such a resource in the coming year. Importantly, we have done our utmost best to provide all the relevant data as supplemental files. These files can either be used as .csv or simply as excel files and as such are easily searchable. We have also included all kinds of additional gene information making it easier for the user to search the literature or link our data to their own favorite strain or organism. We always take care that all of our data is available as raw and processed data and are available to help those that are interested to search or explore our data further. What we presented here is all we can do for now, but we believe it is quite comprehensive.

Reviewer 3. Major comments:

Reviewer 3 - Comment 1.

The overarching goal of the study is to identify novel ways of rescuing drug sensitivity but also identify signals/features that could suggest the emergence of resistance. Overall, while selected examples make a strong case towards the goal, the study does not provide an approach or a tool to predict emergence of resistance.

Author response:

The entire dataset mapped to individual pathways highlights the many opportunities towards lowered ABX sensitivity, a change in relative growth and/or tolerance. We show in Fig 7C that by combining these *in vitro* data with *in vivo* data that we can make predictions on which changes could retain fitness *in vivo* as well as their lowered ABX sensitivity, which we confirm in Fig 7E. Moreover, in Fig.8 we show that changes to genes that our data-set predicts that would trigger lowered drug sensitivity are enriched in clinical samples. Additionally, such

changes are often associated with strains that are resistant to an ABX. This suggest that it could not only be worthwhile to start looking in much more detail into the effects that off target mutations can have on altered drug sensitivity and the emergence of resistance; i.e., such mutations can be the harbingers of the emergence of resistance. In this light, a dataset like ours could thereby be used to guide and make possible predictions on what is important in the emergence of resistance. Our combined dataset and analyses thus provide a pathway towards making predictions. We have clarified these points by making changes throughout the manuscript and especially to the conclusion section.

Reviewer 3 - Comment 2.

The opportunity to use the data for in silico modeling is not implemented or discussed.

Author response:

Implementation of modeling in this manuscript would have been too much, and goes beyond its scope. Such modelling is indeed an excellent idea and we are using our work/data as such in another project. We now explain this more explicitly and reference other examples in the conclusion section on **P15-16**.

Reviewer 3 - Comment 3.

A one-stop list of candidate “precursors” of resistance, and of candidate mediators of persistence, is desired. It is hard to gather from the large amount of supplemental data. The authors should provide a detailed guide to the content of the latter. Along such lines, but conversely, the overview of the study in panel A of Figure 1 is informative.

Author response:

We have done our utmost best to make our data as accessible and explorable as possible in a way that most people can enjoy and benefit from. Highly detailed and explorable data can be found in Supplementary Figures 2 and 3 and Supplementary Table 2, which both visually and numerically show how changes to specific pathways and/or processes lead to antibiotic specific phenotypes. We have also made textual changes to the manuscript to better explain how fitness and altered ABX sensitivity translates into different phenotypes and how this can be extracted from the Figures and Tables. As such we have gone as far as we can possibly go in adding detail and interpretation to the data and manuscript, and thereby making it as accessible as we can. Moreover, as described above, we are in the process of building an online explorable resource, which we hope and aim to complete within the coming months.

Reviewer 3 - Comment 4.

The many Figures presented are beautiful but also really dense and complex. Many undefined abbreviations, acronyms, and truncated words are used throughout, making it hard to intuitively understand the message conveyed. A striking example, that pertains to the figures and the main text, is that only one of the four classes of antibiotics, “CWSI”, is defined. Such terms should be introduced/defined for the general readership.

Author response:

We appreciate the reviewer can see the amount of time and effort we have invested in designing the figures for our manuscript. Importantly, many of the figures incorporate multiple complexities to highlight specific aspects and to enable new discoveries, interpretations, and/or predictions. We agree some figures can be dense at times. We have made textual changes throughout the manuscript to make some of the most difficult figures easier to understand. Additionally, we have gone through the manuscript to make sure all abbreviations and truncated words are properly defined.

Reviewer 3 Minor comments:

1. Line 36: change “generate” to “generated”.
2. Line 170: replace “))” by “)”.
3. Figure 3 a and b: the x-axes should be better defined. How do “effects” compare to “phenotypes” (used in the text) vs. “instances”?

Author response:

We have made all the suggested changes.

REFERENCES

- 1 van Opijnen, T., Bodi, K. L. & Camilli, A. Tn-seq: high-throughput parallel sequencing for fitness and genetic interaction studies in microorganisms. *Nat Methods* **6**, 767-772, doi:10.1038/nmeth.1377 (2009).
- 2 van Opijnen, T. & Camilli, A. Transposon insertion sequencing: a new tool for systems-level analysis of microorganisms. *Nature reviews. Microbiology* **11**, 435-442, doi:10.1038/nrmicro3033 (2013).
- 3 van Opijnen, T. & Levin, H. L. Transposon Insertion Sequencing, a Global Measure of Gene Function. *Annu Rev Genet*, doi:10.1146/annurev-genet-112618-043838 (2020).
- 4 Clark, S. A. *et al.* Topologically correct synthetic reconstruction of pathogen social behavior found during *Yersinia* growth in deep tissue sites. *eLife* **9**, doi:10.7554/eLife.58106 (2020).
- 5 van Opijnen, T. & Camilli, A. A fine scale phenotype-genotype virulence map of a bacterial pathogen. *Genome Res* **22**, 2541-2551, doi:10.1101/gr.137430.112 (2012).
- 6 Mann, B. *et al.* Control of virulence by small RNAs in *Streptococcus pneumoniae*. *PLoS pathogens* **8**, e1002788, doi:10.1371/journal.ppat.1002788 (2012).
- 7 Carter, R. *et al.* Genomic Analyses of Pneumococci from Children with Sickle Cell Disease Expose Host-Specific Bacterial Adaptations and Deficits in Current Interventions. *Cell Host and Microbe* **15**, 587-599, doi:papers://58864D70-D09B-4A16-B641-1A3EEB7FE0BA/Paper/p22252 (2014).
- 8 van Opijnen, T., Dedrick, S. & Bento, J. Strain Dependent Genetic Networks for Antibiotic-Sensitivity in a Bacterial Pathogen with a Large Pan-Genome. *PLoS pathogens* **12**, e1005869, doi:10.1371/journal.ppat.1005869 (2016).
- 9 Jensen, P. A., Zhu, Z. & van Opijnen, T. Antibiotics Disrupt Coordination between Transcriptional and Phenotypic Stress Responses in Pathogenic Bacteria. *Cell Rep* **20**, 1705-1716, doi:10.1016/j.celrep.2017.07.062 (2017).

REVIEWERS' COMMENTS

Reviewer #1 (Remarks to the Author):

The authors have done a good job in addressing the points raised in our original review. We recommend the revised paper for publication in Nature Communications.

Reviewer #2 (Remarks to the Author):

After reviewing the revised manuscript and considering the authors responses to my original critiques, I am satisfied that all of my original concerns have been addressed. It remains my opinion that this is a long and complex paper to get through, but understand that this may be necessary given the scope of the work presented.

I support publication of this paper.